# The structure of a polygamous repressor reveals how phage-inducible chromosomal islands spread in nature

J. Rafael Ciges-Tomas [1,4], Christian Alite[1,4], Suzanne Humphrey [2,4], J. Donderis[1], Janine Bowring[2], Xavier Salvatella [3], José R. Penadés [2] & Alberto Marina [1]

Stl is a master repressor encoded by *Staphylococcus aureus* pathogenicity islands (SaPIs) that maintains integration of these elements in the bacterial chromosome. After infection or induction of a resident helper phage, SaPIs are de-repressed by specific interactions of phage proteins with Stl. SaPIs have evolved a fascinating mechanism to ensure their promiscuous transfer by targeting structurally unrelated proteins performing identically conserved functions for the phage. Here we decipher the molecular mechanism of this elegant strategy by determining the structure of SaPIbov1 Stl alone and in complex with two structurally unrelated dUTPases from different *S. aureus* phages. Remarkably, SaPIbov1 Stl has evolved different domains implicated in DNA and partner recognition specificity. This work presents the solved structure of a SaPI repressor protein and the discovery of a modular repressor that acquires multispecificity through domain recruiting. Our results establish the mechanism that allows widespread dissemination of SaPIs in nature.

[1] Instituto de Biomedicina de Valencia (IBV-CSIC) and CIBER de Enfermedades Raras (CIBERER), Valencia 46010, Spain. [2] Institute of Infection, Immunity and Inflammation, College of Medical, Veterinary and Life Sciences, University of Glasgow, Glasgow G12 8TA, UK. [3] ICREA and Institute for Research in Biomedicine (IRB Barcelona), The Barcelona Institute of Science and Technology, Barcelona 08010, Spain. [4] These authors contributed equally: J. Rafael Ciges-Tomas, Christian Alite, Suzanne Humphrey. Correspondence and requests for materials should be addressed to J.R.P. (email: JoseR.Penades@glasgow.ac.uk) or to A.M. (email: amarina@ibv.csic.es)

The *Staphylococcus aureus* pathogenicity islands (SaPIs) are the prototypical members of the phage-inducible chromosomal island (PICI) family of mobile genetic elements[1]. They are very widespread among the Staphylococci and are responsible for at least one important human disease: toxic shock syndrome[1]. These islands are innately tied to the bacteriophage life cycle, requiring a helper phage for their induction and transfer. In the absence of a helper phage they reside stably in the host chromosome under control of the SaPI-encoded master repressor, Stl[2]. Stl is a DNA-binding protein that binds to the intergenic promoter region between *stl* and *str*[3] (the SaPI transcription rightward regulator), preventing transcription of the *str* and downstream genes, thus maintaining integration of the island in the host chromosome. SaPI derepression occurs following a direct interaction between the Stl repressor and a specific phage-encoded inducer protein[3], which disrupts the Stl–DNA complex, leading to activation of the SaPI cycle. Different SaPIs encode different Stl repressor proteins, which are highly divergent in sequence, suggesting that each SaPI requires a different phage inducer protein for island induction. Hence, the islands SaPIbov1, SaPIbov2, SaPI1 and SaPI2 are induced by dUTPase (Dut), φ80α ORF15, Sri and recombinase phage proteins, respectively[3–5].

Although it was initially thought that each of the different Stl repressors could uniquely interact with an individual phage-encoded protein, recent work by our laboratories has identified that, instead of targeting a specific phage protein for their derepression, the SaPI-encoded Stl repressors can target multiple phage inducer proteins that perform the same function for the phages, but have completely divergent structures[4]. For instance, the SaPI2 Stl can target four sequence- and structurally-unrelated families of phage recombinase for SaPI2 induction: Sak, Sak4, Erf, and Redβ recombinases[4,6]. Likewise, the SaPIbov1 Stl can target both trimeric and dimeric Duts encoded by different phages to initiate SaPI replication, despite the radically different structures of the two Dut types[7–10] (Supplementary Note 1, Supplementary Fig. 1).

As with the SaPIs, all the characterized PICIs encode a functionally related Stl-like repressor, Rpr[11,12]. Functionally, the Rpr proteins resemble the cI/Cro family of repressors found in temperate phages. Both types of repressors prevent excision and replication of the mobile element (PICIs and phages, respectively) by binding to specific regions through an N-terminal HTH DNA-binding domain[1,13]. Canonical repressors of the cI/Cro family link to this HTH domain a C-terminal domain, of reduced size in several cases, that promotes dimerization[14]. Induction of the genes under the repression of these regulators should involve the disruption of the dimeric organization by the interaction with a derepressor protein or by the direct cleavage between the N- and C-terminal domains mediated by proteases in the bacterial host cell (such as RecA*, induced by the SOS response)[15,16].

How does this happen? No Stl repressor structure has been solved. However, previous studies with the SaPIbov1 Stl repressor proposed an architecture with an N-terminal DNA-binding domain and a C-terminal portion of unknown function that seems to consist of two domains connected by a low complexity segment[4,17], suggesting a potential capacity for Stl proteins to interact with unrelated proteins. We demonstrated that the deletion of the C-terminal portion after the low complexity segment generates a SaPIbov1 Stl that retains the capacity to interact with trimeric but not dimeric Duts, while the deletion of the N-terminal DNA-binding domain has the opposite effect, precluding Stl binding to trimeric but not to dimeric Duts[7]. These results suggested that Stl is composed of different domains with alternative and complementary functional characteristics. Thus, it is intriguing to understand the structural basis for how the SaPIbov1 Stl has acquired the ability to interact with two unrelated families of phage-encoded Duts.

Other unsolved questions include (i) what is the mechanism by which the inducer proteins alleviate the Stl-mediated repression; and (ii) for a specific repressor, is this mechanism conserved among the different inducers? Here, we show the solved structure of the SaPIbov1 Stl repressor. Furthermore, by solving the structure of the SaPIbov1 Stl complexed with a phage-encoded trimeric or dimeric Dut, we have identified specific domains and residues that are vital to these interactions. This highlights the Stl repressor as a modular protein capable of targeting multiple partners through domain recruiting, and establishes the molecular mechanism by which the clinically important SaPIs pirate conserved phage mechanisms to spread in nature.

## Results

**Crystal structures confirm Stl modularity**. To better understand the mechanism of action of the Stl repressor, we set out to determine the crystal structure of SaPIbov1 Stl (BovI-Stl) repressor but our attempts were unsuccessful. We hypothesized that the proposed low complexity region in BovI-Stl could confer high flexibility to the molecule, precluding crystallization of the whole Stl. Therefore, we determined to solve the structure of BovI-Stl by eliminating this region and dividing the protein into two functional and complementary portions corresponding to the N-terminal (BovI-Stl$^{N-ter}$; residues 1–156) and to the C-terminal part of the protein (BovI-Stl$^{C-ter}$; residues 175–267) (Fig. 1a). We confirmed by Native-PAGE that BovI-Stl$^{N-ter}$ and BovI-Stl$^{C-ter}$ preserve the capacity to interact with trimeric and dimeric Duts, respectively (Supplementary Fig. 2). Using this strategy, we successfully solved the crystal structures of both BovI-Stl$^{N-ter}$ and BovI-Stl$^{C-ter}$, Stl fragments.

The crystal structure of BovI-Stl$^{N-ter}$ was solved to 1.8 Å resolution and showed a single copy of the protein in the asymmetric unit (Table 1) that presents an α-helical folding, as was anticipated by in silico modelling, composed of 10 α-helices (Figs. 1a,b). The initial four α-helices (α1–α4) correspond to the DNA-binding domain, presenting the archetypal helix-turn-helix (HTH) motif (Fig. 1b) observed in multiple repressors, as well other DNA-binding proteins[18] (Supplementary Fig. 3a). Helix α5 connects the HTH domain with the remaining five helices that form a more compact helical bundle. We named this portion of the protein the middle domain due to its localization in the Stl sequence (Fig. 1a, b). A search for overall structural similarities with BovI-Stl$^{N-ter}$ using the programme DALI[19] clearly identified matches for the HTH domain but failed to reveal any significant match for the middle domain.

Likewise, we solved the structure of BovI-Stl$^{C-ter}$ to 2.2 Å resolution. BovI-Stl$^{C-ter}$ is also an α-helical domain forming an antiparallel three-helix bundle (Fig. 1c). We have named these helices as α11, α12 and α13 to follow the BovI-Stl$^{N-ter}$ numbering (Fig. 1a, c). Nine residues (230–238) that correspond to a loop connecting α11–α12 were disordered, as well the three initial residues and the last one. The asymmetric unit of the crystal contains a single copy of BovI-Stl$^{C-ter}$ but the analysis of probable assemblies by PISA software[20] proposed a dimeric organization by exploiting a two-fold crystallographic axis with a complex formation significance score of 1 (highest score). The dimerization is mediated by the reciprocal interaction of the N-terminal part of helices α11 and α13, expanding the bundle from three to six helices (Fig. 1c, Supplementary Table 1). Around 800 Å$^2$ of total surface area would be buried per monomer upon BovI-Stl$^{C-ter}$ dimerization, suggesting a stable dimer that was confirmed by size exclusion chromatography (SEC) analysis (Supplementary Fig. 4). Mixed hydrophobic and hydrophilic interactions, including a double salt bridge between E189 and R253, are present in the dimer interface (Supplementary Table 1).

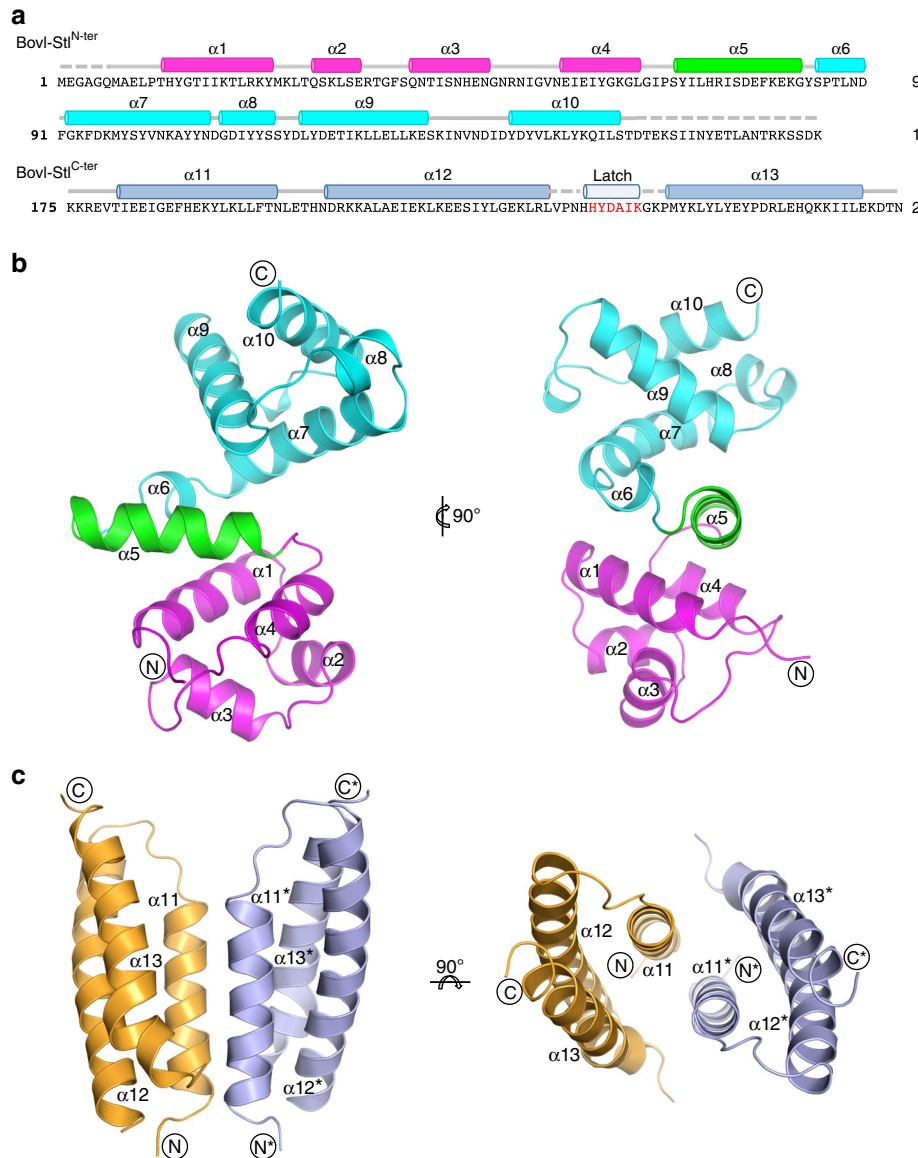

**Fig. 1** Structure of SaPIbov1 Stl domains. **a** Sequence of SaPIbov1 Stl. The sequence of SaPIbov1 Stl is divided as BovI-Stl[N-ter] and BovI-Stl[C-ter] corresponding to the fragments used to solve the crystal structures. Structural elements are shown above the sequence coloured as **b** and **c**. The latch observed in the BovI-Stl[C-ter] structure in complex with φO11 Dut is indicated in red. **b** Cartoon representation of SaPIbov1 Stl[N-ter]. Two orthogonal views of the BovI-Stl[N-ter] structure are shown and the secondary structural elements are numbered and labelled in order from the N to C terminus. The HTH domain is coloured in magenta and the middle domain in cyan, with the connecting α5 helix in green. **c** Structure of SaPIbov1 Stl[C-ter]. Cartoon representation of the BovI-Stl[C-ter] dimer with protomers coloured in blue and orange, respectively. Two orthogonal views are shown and the secondary structural elements are labelled following the numbering of BovI-Stl[N-ter] (the asterisk indicates elements from the second protomer)

**DNA recognition by Stl.** According to the DALI server[19], the BovI-Stl HTH domain superimposes well with the corresponding domains of different DNA-binding proteins, in particular with the bacterial restriction-modification (R-M) controller protein from the Esp1396I system (C.Esp1396I) that allowed us to generate a plausible biological model of the Stl dimer bound to DNA (Fig. 2, Supplementary Fig. 3, Supplementary Note 3).

In the model, BovI-Stl recognizes the DNA by inserting helix α3 of the HTH motif into the major groove of DNA (Fig. 2a, b). Residues Q40, N41, T42, S44, N45 and R51 would be well placed to read-out the TATCTC nucleotide sequence recognized by Stl[21,22], making specific interactions through the major groove (Fig. 2b, Supplementary Fig. 5). A BovI-Stl Q40-N41 to A40-A41 double mutant was previously shown to be highly defective in DNA binding based on electrophoretic mobility shift assay

(EMSA) experiments[17], supporting our model of DNA recognition. Our model further proposes that additional residues such as Q29, S44, N48 and R22 could mediate the nonspecific read-out of the nucleotide sequence by interacting with the DNA backbone (Supplementary Fig. 5). The first three residues coordinate a sulfate ion in the BovI-Stl[N-ter] structure that could mimic a DNA phosphate, since superimposition with the C.Esp1396I-DNA complex places this sulfate at the position of a DNA backbone phosphate (Fig. 2b).

To experimentally confirm our model, we generated single BovI-Stl mutations to alanine of residues predicted to be responsible for specific and nonspecific DNA read-out, R22 (Stl[R22A]), Q29 (Stl[Q29A]), S44 (Stl[S44A]), N45 (Stl[N45A]), N48 (Stl[N48A]) and R51 (Stl[R51A]), as well as E47 (Stl[E47A]), which could interact with DNA backbone recognition residues R22 and Q29.

**Table 1 Data collection and refinement statistics**

|  | BovI-Stl$^{C-ter}$ | Dutφ011-BovI-Stl$^{C-ter}$ | BovI-Stl$^{N-ter}$ | Dutφ11-BovI-Stl$^{N-ter}$ |
|---|---|---|---|---|
| **Data collection** |  |  |  |  |
| Beamline | ALBA-XALOC | ALBA-XALOC | DLS I0-3 | DLS I-04 |
| Wavelength (Å) | 0.97906 | 0.97926 | 0.97623 | 0.9282 |
| Space group | P321 | I23 | C2 | P321 |
| Cell dimensions (Å) | $a = b = 77.4\ c = 37.3$ | $a = b = c = 122.9$ | $a = 132.5\ b = 34.6\ c = 37.0$ | $a = b = 144.5\ c = 149.4$ |
|  | $\alpha = \beta = 90\ \gamma = 120$ | $\alpha = \beta = \gamma = 90$ | $\alpha = \gamma = 90\ \beta = 95.96$ | $\alpha = \beta = 90\ \gamma = 120$ |
| Resolution (Å)[a] | 67.0–2.2 | 86.9–2.9 | 33.6–1.8 | 95.9–2.52 |
|  | (2.32–2.2) | (3.08–2.9) | (1.83–1.8) | (2.56–2.52) |
| Total reflections | 1,119,769 (14,043) | 232,953 (37,049) | 100,090 (4685) | 649,186 (27,668) |
| Unique reflections | 6522 (843) | 7002 (1110) | 15,780 (777) | 61,269 (3036) |
| Completeness (%) | 96.7 (88.3) | 100 (100) | 100 (99.4) | 100 (100) |
| Multiplicity | 18.4 (16.7) | 33.3 (33.4) | 6.3 (6.0) | 10.6 (9.1) |
| Mean $I/\sigma(I)$ | 14.4 (2.8) | 23.7 (4.8) | 12.4 (9.7) | 12 (1.2) |
| Rpim | 0.027 (0.253) | 0.018 (0.183) | 0.09 (0.574) | 0.052 (0.745) |
| CC 1/2 | 0.999 (0.933) | 1 (0.913) | 0.985 (0.568) | 0.999 (0.477) |
| **Refinement** |  |  |  |  |
| $R_{work}$ | 0.248 | 0.253 | 0.172 | 0.197 |
| $R_{free}$ | 0.263 | 0.305 | 0.216 | 0.240 |
| Number of atoms | 684 | 1727 | 1303 | 9321 |
| Protein | 665 | 1709 | 1208 | 9072 |
| Water | 19 | 18 | 85 | 193 |
| Others | - | - | 10[b] | 56[c] |
| Rmsd, bonds (Å) | 0.008 | 0.009 | 0.0183 | 0.0147 |
| Rmsd, angles (°) | 1.10 | 1.50 | 1.782 | 1.734 |
| MolProbity Clashscore | 6.69 | 21.18 | 2.92 | 3.34 |
| (Percentile) | (98th) | (91st) | (99th) | (100th) |
| **Ramachandran plot** |  |  |  |  |
| Preferred (%) | 97.626 | 94.2 | 98.6 | 98.5 |
| Allowed (%) | 2.74 | 5.8 | 1.4 | 1.5 |
| Outliers (%) | 0 | 0 | 0 | 0 |
| **PDB ID codes** | 6H48 | 6H4B | 6H49 | 6H4C |

[a]Number in parentheses indicate values for the highest-resolution cell
[b] Atoms corresponds to two sulfate molecules
[c]Atoms correspond to two Mg ions, five Ni ions, and seven di-(hydroxyethyl)-ether molecules

The different BovI-Stl mutations were separately introduced into the plasmid pJP2085, which carries a β-lactamase reporter gene fused to *xis*, downstream of *str* and the BovI-Stl-repressed *str* promoter, and also encodes BovI-Stl (Fig. 2c). These plasmids were introduced into the non-lysogenic *S. aureus* RN4220 strain, where induction of *xis* expression (normally repressed by BovI-Stl) was used as a proxy for island induction. All the BovI-Stl mutants were defective in SaPI repression, indicating uncontrolled island activation in the absence of helper phage (Fig. 2c). To confirm that these BovI-Stl mutations impair binding to the SaPI promoter, they were recombinantly produced in *E. coli* and EMSA experiments using the SaPIbov1 *stl* and *str* promoter region were performed. All the BovI-Stl mutants presented a reduced or null capacity for DNA binding when compared with the wild-type protein, in agreement with the in vivo experiments (Fig. 2d). Mutations in residues not predicted to interact with DNA had no effect on DNA binding and conserved island repression capacity (see below). SEC-MALS and thermofluor experiments confirmed that these mutants form dimers in solution and have a similar denaturalization temperature ($T_m$) range as the WT protein, ruling out that the functional deficiencies observed were due to structural side effects induced by the mutations (Supplementary Table 2).

**Structure of BovI-Stl$^{N-ter}$ bound to trimeric φ11 Dut**. The structure of BovI-Stl$^{N-ter}$ in complex with the trimeric Dut of phage φ11 (Dutφ11) was solved to 2.52 Å resolution by molecular replacement using the structure of Dutφ11 as a model (PDB

4GV8; see ref. [23], Table 1). The structure showed that the Dut maintains its trimeric architecture and interacts with three independent BovI-Stl$^{N-ter}$ monomers (Fig. 3a), confirming the 1:1 stoichiometry previously proposed for the Stl-trimeric Dut complex[4,24]. Comparison of the BovI-Stl$^{N-ter}$ Dutφ11 complex with the structures of both proteins alone showed that complex formation induces modest changes (Supplementary Note 2), mainly in Stl (Supplementary Fig. 6), indicating that their structural conformations in solution are competent for interaction. BovI-Stl$^{N-ter}$ binds primarily to Dutφ11 through interactions with residues conforming the Dut active centre (Fig. 3, Supplementary Table 3), precluding the binding of the nucleotide and the consequent motif V positioning. This mode of interaction explains both the Stl-mediated inhibition of dUTPase activity of the phage trimeric Duts, and the blocking of Stl binding to the Dut by the substrate dUTP[24–26].

Further examination of the Stl–Dutφ11 interaction showed that BovI-Stl inserts the helix α8 and the α8–α9 connecting loop (Lα8α9) into the Dutφ11 active centre, mimicking the interactions mediated by the substrate with catalytic residues on the conserved motifs of trimeric Duts (Fig. 3b, c; Supplementary Tables 3 and 4). Thus, Stl residues Y112 and Y113 place at the positions occupied by the nucleotide ribose and pyrimidine ring, respectively, with Y112 interacting with the catalytic residues D81 and Y84 on the conserved motif III (Fig. 3b, c; Supplementary Tables 3 and 4). Stl residue Y105 is placed in the position occupied by the dUTP γ-P and residue Y106 is hydrogen bound with the side-chains of Dutφ11 residues H21 and D24 at the beginning of motif I (Fig. 3b, c; Supplementary Tables 3 and 4).

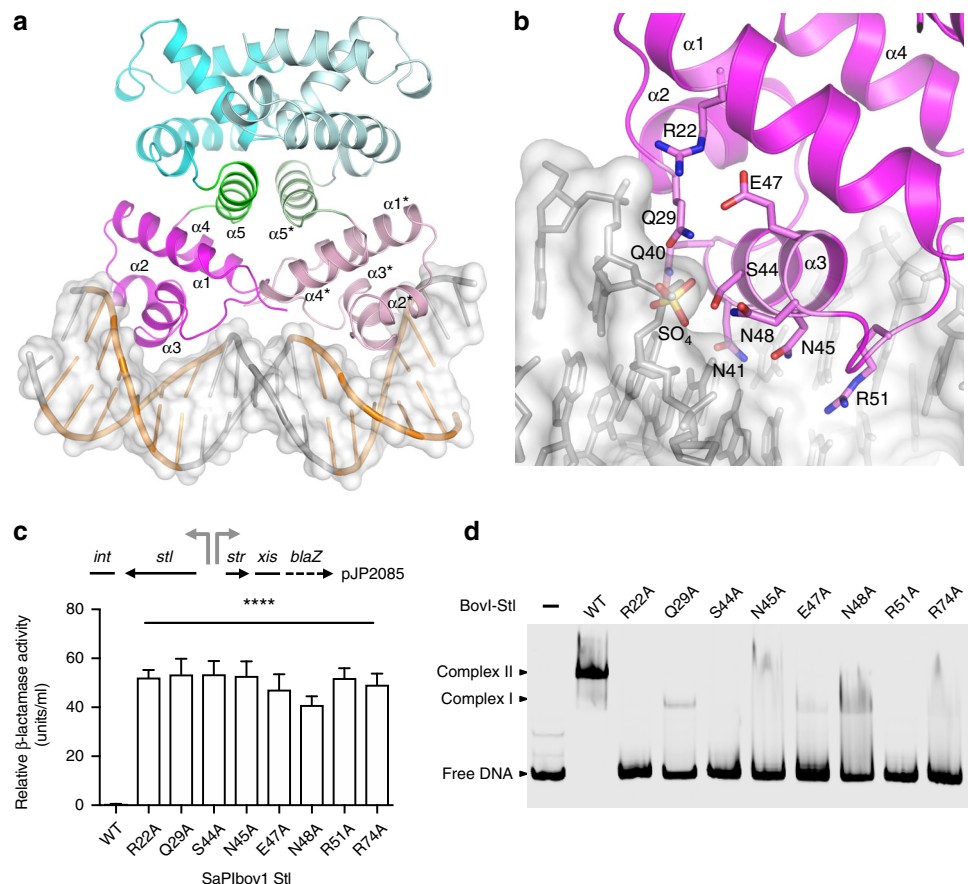

**Fig. 2** Structural and functional analysis of the DNA-binding model of the Stl dimer. **a** Model of dimeric Stl bound to the palindromic SaPIbov1 repression site. The BovI-Stl[N-ter] dimer is presented in cartoon representation with one protomer coloured as in Fig. 1b and the other with lighter tones. DNA is shown in grey sticks and white surface with the TATCTC palindromic sequences highlighted in orange. Binding of the dimer is based on the superimposition of individual BovI-Stl[N-ter] protomers with the C.Esp1396I complex and its palindromic DNA operator (PDB:3CLC[41]; Supplementary Fig. 2b). Structural elements of the HTH DNA-binding domain and the connector helix are labelled. **b** Close view of the modelled HTH-DNA interaction. Stl DNA interaction residues are highlighted in stick, coloured by atom type and labelled. The sulfate ion that is presented in the BovI-Stl[N-ter] structure is also shown in stick and labelled, and superimposes on a DNA backbone phosphate. The HTH α-helices are labelled. **c** Unregulated *str* transcription due to SaPIbov1 Stl repressor DNA- binding domain mutations. Top left: schematic of a *blaZ* transcriptional fusion in pJP2085. β-Lactamase assays were performed with RN4220 containing the pJP2085 SaPIbov1 Stl[WT] or derivatives, with data from three independent experiments. Error bars represent SD. ANOVA with Tukey's multiple comparisons test compared mean differences between the Stl[WT] control and the mutants. Significant adjusted *p* values relative to the WT were: Stl[R22A] < 0.0001[****]; Stl[Q29A] < 0.0001[****]; Stl[S44A] < 0.0001[****]; Stl[N45A] < 0.0001[****]; Stl[E47A] < 0.0001[****]; Stl[N48A] < 0.0001[****]; Stl[R51A] < 0.0001[****]; Stl[R74A] < 0.0001[****]. **d** In vitro evaluation of mutations in the Stl DNA-binding domain. EMSA gel results of Stl[WT] and mutants in the residues predicted to interact with DNA in the model of panel **b**. In the presence of Stl[WT] the band corresponding to the *stl-str* DNA operator region, labelled as free DNA, disappears and two new bands corresponding to Stl-DNA complexes are observed (labelled as complex I and II; see ref. [21]). For the Stl mutants the bands corresponding to Stl-DNA complexes almost completely disappear while most of the DNA appears at the lower position of the gel as free DNA. One representative experiment of at least three independent replicates is shown. Source data are provided as a Source Data file

The Dutϕ11 H21 residue also interacts with Stl residue N102 (Fig. 3b; Supplementary Table 3). In the opposite part of the active centre, Stl residue Y116 in Lα8α9 is inserted between the Dut conserved catalytic motif IV and the phage-specific motif VI, interacting with Dutϕ11 K131 from the former motif and with I110 from the second (Fig. 3b; Supplementary Table 3). The main-chain of Stl Y116 also contacts the Dutϕ11 R64, a key catalytic residue from the motif II (Fig. 3b; Supplementary Table 3). The D117 residue in the Stl Lα8α9 forms a further interaction through a salt-bridge with the Dutϕ11 motif IV residue K131 (Fig. 3b; Supplementary Table 3). Consequently, this short region of Stl occupies the Dut active centre and contacts residues from all the conserved catalytic motifs with the exception of motif V, which remains disordered in the Dutϕ11–Stl complex. In addition, the R74 and D77 residues from Stl helix α5, which connect the HTH and middle domain, are also involved in

Stl–Dut recognition, forming a secondary, albeit more modest, area of interaction. In this area of interaction, residues R74 and D77 form salt bridges with Dutϕ11 residues preceding the Dut motif I, E18 and R15, respectively (Fig. 3b; Supplementary Table 3).

**Stl[N-ter] binds trimeric Dut by mimicking dUTP interactions.** We have previously characterized several mutations in Dutϕ80α and Dutϕ11 involving residues from the conserved catalytic motifs (Supplementary Fig. 1), which significantly affect the Stl–Dut interaction. These included the mutation of the catalytic D81 and Y84 to different residues that in vivo completely hamper or strongly reduce the island induction for Dutϕ80α or Dutϕ11, respectively[5,26], with subsequent in vitro work confirming their pivotal role in driving recognition between both proteins[5,26]. In

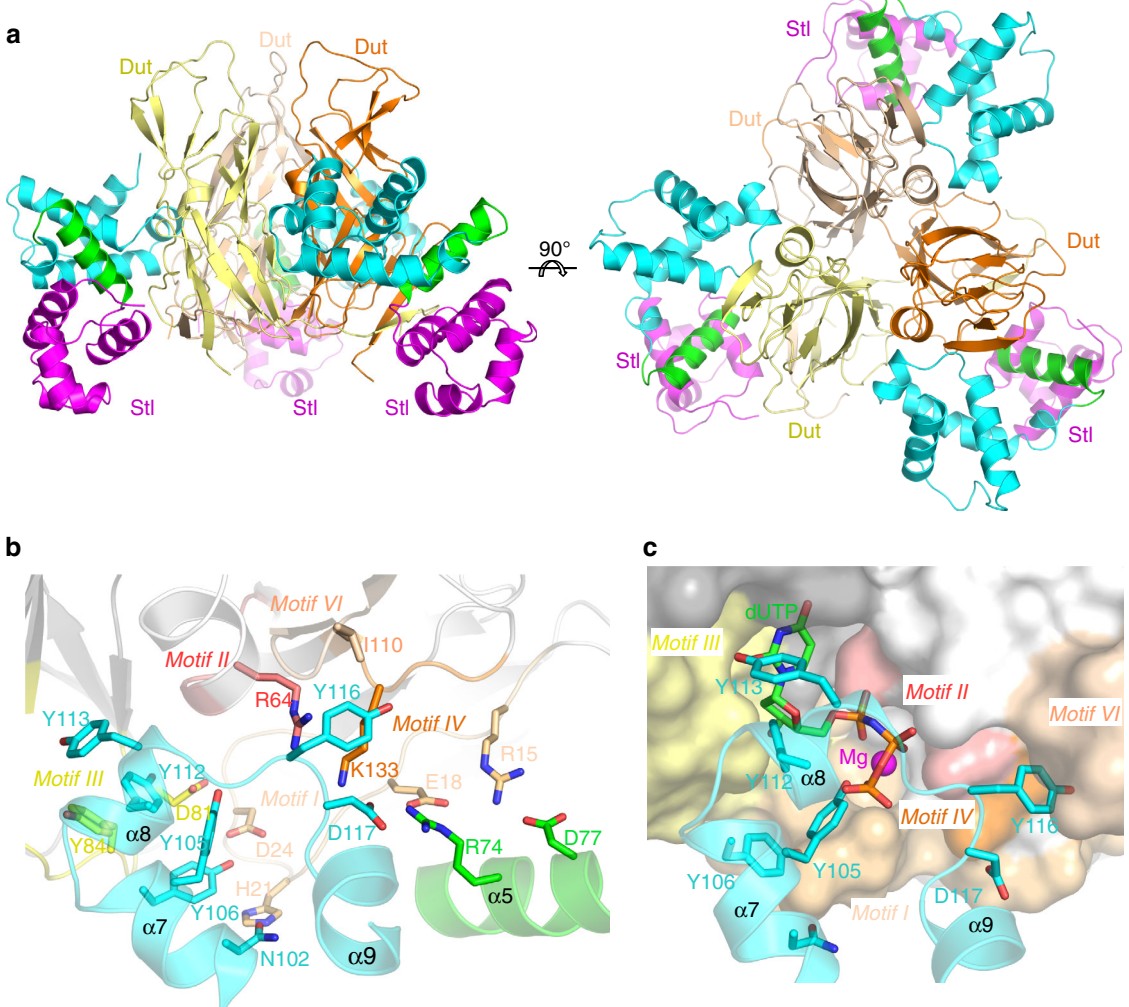

**Fig. 3** Crystal structure of the BovI-Stl^N-ter-Dut φ11 complex. **a** Structure of the complex between the trimeric Dut from phage φ11 (protomers coloured in different tones of yellow-orange) and three molecules of BovI-Stl^N-ter (protomers coloured as in Fig. 1b). Two orthogonal views are shown. **b** Close-up view of the interaction of a protomer of BovI-Stl^N-ter with the Dutφ11 trimer. Notice that only two of the three protomers of Dutφ11 contact each Stl molecule. Dut catalytic motifs are highlighted in different colours and labelled. The residues involved in interactions are shown as sticks, labelled and coloured by atom type with the carbons in the same colour as the corresponding molecule. For clarity, only the structural elements of BovI-Stl^N-ter involved in the interaction are shown and labelled. **c** Stl mimics the interactions of the dUTP substrate with Dutφ11. The substrate dUTP was placed in the active centre of Dutφ11 in complex with BovI-Stl^N-ter by superimposing the structure of Dutφ11-dUTP (PDB 4GV8) and is shown as sticks with carbon atoms in green. The Mg ion chelated by the dUTP is shown as a magenta sphere. Dutφ11 is represented in surface and BovI-Stl^N-ter in cartoon and sticks. Structural elements and BovI-Stl^N-ter interacting residues are labelled as well as conserved catalytic motifs of Duts and coloured as in **b**. The omit electron density map is shown in Supplementary Fig. 13. Source data are provided as a Source Data file

addition, we previously showed that three residues in motif IV modulate Dut–Stl affinity, with the residue combination present in Dutφ11 (132-DKL-134) having a higher affinity for the Stl than the Dutφ80α combination (133-ERI-135)[26]. The Dut–Stl complex shown here reveals that Dutφ11 interacts with the Stl through residue K133 (Fig. 3b; Supplementary Table 3), which would correspond to the Dutφ80α R134, implying that the change in this residue contributes to this difference in affinity.

To evaluate the contributions of the counterpart BovI-Stl residues to complex formation with trimeric Duts, the effect of mutating some of the BovI-Stl interacting residues was tested in vivo and in vitro with both Dutφ11 and Dutφ80α trimeric Duts, as well with the DutφNM1 dimeric Dut to evaluate the contribution of these positions in Dut-type selection. Mutation to alanine of Stl residues Y106 (Stl^Y106A), Y112 (Stl^Y112A), Y113 (Stl^Y113A) and Y116 (Stl^Y116A) generated repressors that were unable to interact with the trimeric Dutφ80α but that interacted

even better with the dimeric DutφNM1 (Fig. 4a). A possible explanation for this increased induction could be related with the fact that these residues are important to stabilize the BovI-Stl dimer. The results with the trimeric Dutφ11 were intriguing, since only the Stl^Y112A showed a reduced capacity to interact with this Dut (Fig. 4a). Since we had previously demonstrated that the Dutφ11 has more affinity for BovI-Stl than Dutφ80α[26,27], we hypothesized that a single mutation would be not enough to disrupt the Dutφ11–BovI-Stl interaction. Thus, we generated a BovI-Stl carrying the double Y112A/Y113A (Stl^YY-AA) mutation and tested its ability to interact with the different Duts. In support of our hypothesis, the Stl^YY-AA repressor maintained its ability to repress the island and be induced by the dimeric DutφNM1, but completely lost the ability to be induced by trimeric Duts (Fig. 4a).

The residue mutations Stl^Y112A and Stl^YY-AA were recombined into strains containing the SaPIbov1 island for further testing

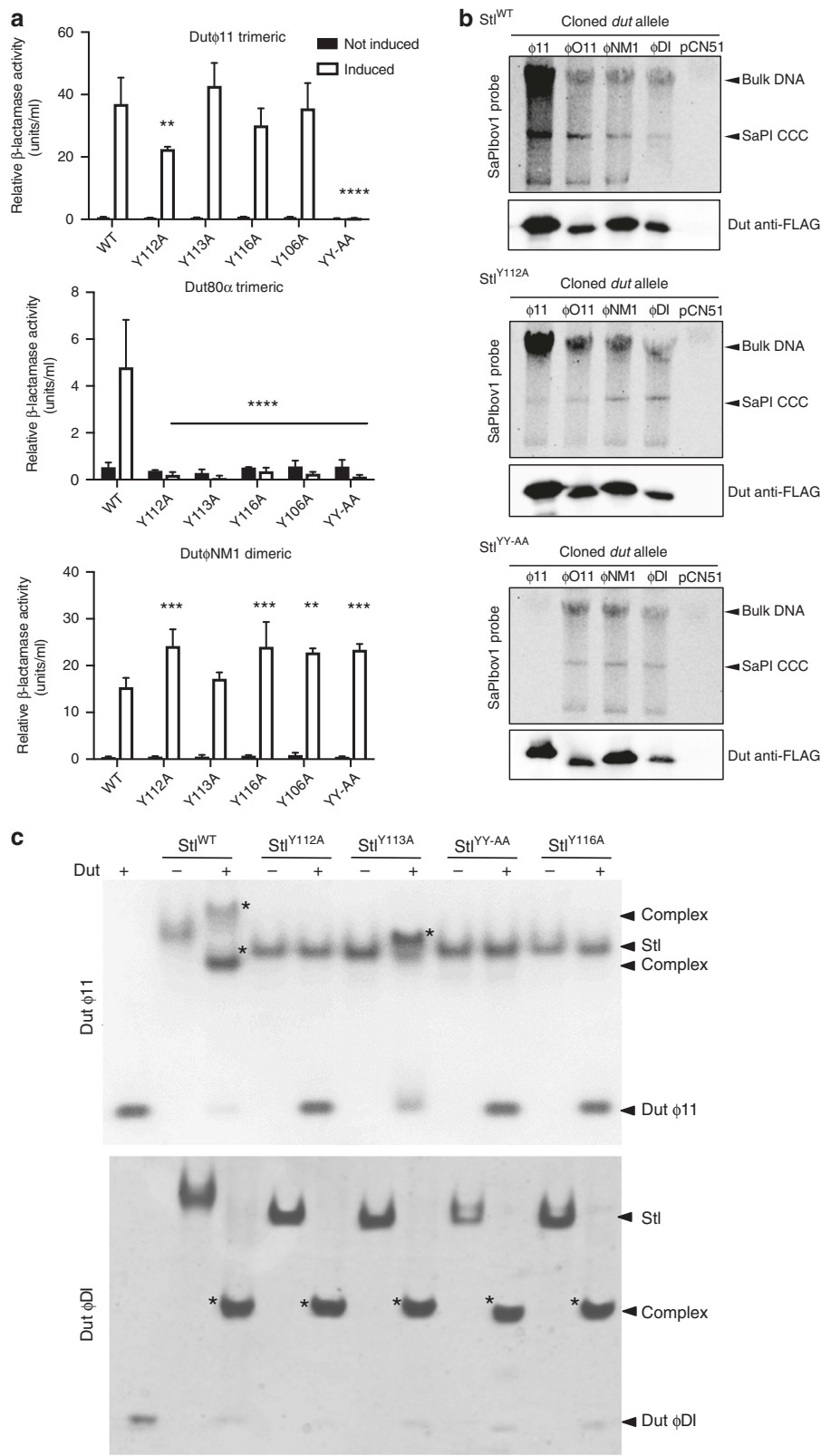

together with either Duts encoded on the cadmium-inducible overexpression vector pCN51 or with the phages encoding specific Dut genes (see Methods for more details). When the trimeric Dutφ11 was overexpressed from the *P*cad promoter on pCN51 in the Stl$^{YY-AA}$ mutant strain, SaPIbov1 induction was extraordinarily reduced in comparison to the WT SaPIbov1 island (Fig. 4b). Interestingly, the trimeric Dutφ11 was able to

induce the island carrying the Stl$^{Y112A}$ mutation, although was clearly reduced compared that for the WT SaPIbov1. This can be explained since the phage φ11 is still able to de-repress the island (Fig. 4a) with the concomitant expression of the SaPIbov1 *pri-rep* genes involved in SaPI replication. When dimeric Duts DutφDI, DutφNM1 and DutφO11 were expressed SaPIbov1 induction in the mutant strains was equivalent to the WT, indicating that the

**Fig. 4** In vivo and in vitro evaluation of trimeric Dut recognition and binding by Stl. **a** Mutations Stl$^{Y112A}$ and Stl$^{YY-AA}$ reduce or abolish Stl interaction with the trimeric Duts, but not the dimeric Duts. β-Lactamase assays were performed with strains containing the pJP2085 SaPIbov1 Stl$^{WT}$ or derivatives, and lysogenic for either φ11, φ80α or φNM1 encoding for a trimeric or dimeric Dut. Samples were taken after 90 min in the absence or following phage induction with Mitomycin C. All data are the result of three independent experiments. Error bars represent SD. A two-way ANOVA with Tukey's multiple comparisons test compared mean differences between the Stl$^{WT}$ control and the mutants, within columns. Significant adjusted $p$ values were as follows: φ11 induced Stl$^{Y112A}$ 0.0046**, induced Stl$^{YY-AA}$ < 0.0001****; φ80α induced Stl$^{Y112A}$ < 0.0001****, induced Stl$^{Y113A}$ < 0.0001****, induced Stl$^{Y116A}$ < 0.0001****, induced Stl$^{Y106A}$ < 0.0001****, induced Stl$^{YY-AA}$ < 0.0001****; φNM1 induced Stl$^{Y112A}$ 0.0003***, induced Stl$^{Y116A}$ 0.0003***, induced Stl$^{Y106A}$ 0.0020**, induced Stl$^{YY-AA}$ 0.0008***, all other values were not significant. **b** SaPIbov1 Stl$^{WT}$ and Stl$^{Y112A}$ or Stl$^{YY-AA}$ island excision and replication following induction of cloned Dut genes. Strains JP6774, JP17043 and JP17706 containing SaPIbov1 with Stl$^{WT}$, Stl$^{Y112A}$ and Stl$^{YY-AA}$, respectively, were complemented with plasmids expressing either the 3xFLAG-tagged φ11 trimeric Dut or φO11, φNM1 or φDI dimeric Duts. Samples were isolated 3 h post-induction with 1 μM CdCl$_2$ and Southern blots were performed using a SaPIbov1 integrase probe. The upper band is "bulk" DNA, including chromosomal, phage, and replicating SaPI. CCC indicates covalently closed circular SaPI DNA. The lower panels below each Southern are western blots probed with antibody to the FLAG-tag carried by the Dut proteins. **c** Native gel mobility shift assays tested the binding capacity of wild type and mutant Stl proteins to trimeric Dutφ11 and dimeric DutφDI. The appearance of bands with alternated migration with respect to the individual proteins (labelled by asterisk) indicates formation of Stl–Dut complex. Source data are provided as a Source Data file

Stl$^{Y112A}$ and Stl$^{YY-AA}$ mutations impact the Stl interaction exclusively with trimeric Duts (Fig. 4b). Western blotting showed that each Dut expressed similarly in the presence of the different Stl clones, indicating that the differences seen were due to the Stl mutations (Fig. 4b). The different dimeric Duts used here all induce SaPIbov1, but have a variable domain VI that is different in each Dut[7], although this does not seem to impact on Stl interaction capabilities.

The differences between the trimeric and dimeric Dut interactions with the SaPIbov1 mutants were further confirmed using a phage-interference spot test assay (see Methods for explanation) where different strains containing WT or mutant SaPIbov1 (Stl$^{WT}$, Stl$^{Y112A}$ or Stl$^{YY-AA}$) were used as recipients for infection with phages φ11 (trimeric Dut), 80α (trimeric Dut) or φNM1 (dimeric Dut). SaPIbov1 induction is indicated by a reduction in phage titre relative to the no island control. The trimeric Dut-encoding phage φ11 together with the mutant SaPIbov1 island Stl$^{YY-AA}$ did not show the reduced phage titre it exhibits in the presence of WT or Stl$^{Y112A}$ SaPIbov1, indicating there was no SaPI interference due lack of SaPI induction (Supplementary Fig. 7). Similarly, no SaPI interference was detected for trimeric Dut-encoding phage 80α in the presence of either Stl$^{Y112A}$ or Stl$^{YY-AA}$, indicating that SaPI induction is impaired in these mutants. The dimeric Dut encoding φNM1 showed reduced phage titre levels with both WT and mutant SaPIbov1 islands, indicating SaPI induction and interference (Supplementary Fig. 7). We could not analyse the titres of phages φO11 and φDI because they do not produce clear plaques in the recipient strain RN4220.

Finally, to validate these results in vivo, we analysed both the replication and transfer of the WT and SaPIbov1 mutants (Stl$^{WT}$, Stl$^{Y112A}$ or Stl$^{YY-AA}$) by the different phages under study. To do this, the strains carrying the different islands were lysogenized with phages φ11, 80α or φNM1. Samples of each strain were obtained at 90 min post-induction with mitomycin C (activating the phage cycle) to examine island replication by Southern blot, while lysates of each strain were used to quantify the transfer of the different islands. In support of the previous results, the replication and transfer of the SaPIbov1 Stl$^{YY-AA}$ mutant by the phages encoding trimeric Duts (φ11, 80α) was significantly reduced compared to that observed for the WT island. Note that the transfer obtained is similar to that observed with the non-inducing phage carrying the Δ*dut* mutations. As expected for its higher affinity for the SaPIbov1 repressor, the phage φ11 but not the 80α was able to induce and transfer the SaPIbov1 Stl$^{Y112A}$ island (Supplementary Table 5, Supplementary Fig. 8). Also in accordance with the previous results, the dimeric Dut-encoding phage (φNM1) induced and

mobilized all the islands efficiently (Supplementary Table 5, Supplementary Fig. 8).

In vitro characterization of the complex formation using Native-PAGE showed that the Stl$^{Y112A}$, Stl$^{Y113}$, Stl$^{Y116A}$ and Stl$^{YY-AA}$ mutants have reduced or null capacity to interact with the trimeric Dutφ11 (Fig. 4c). Conversely, none of these mutations affected the ability of Stl to complex with the φDI dimeric Dut (Fig. 4c). These two Duts were selected as representatives of trimeric and dimeric Duts since their complexes with Stl are easily appreciable by Native-PAGE. SEC-MALS and thermofluor experiments showed that these mutations have no effect on the oligomerization state and stabilization of BovI-Stl (Supplementary Table 2). These results confirm that Stl is anchored to the trimeric Duts by multiple sites involving catalytic motif residues but, also, that these recognition sites must differ to those involved in dimeric Dut interaction. In addition, the in vitro and in vivo data point to the mimicry of the adenosine motif interactions by Stl residues Y112 and Y113 as the main mechanism by which Stl recognizes trimeric Duts.

**Co-crystal structure of Stl- dimeric φO11 Dut.** The structure of BovI-Stl$^{C-ter}$ in complex with the Dut of phage φO11 (DutφO11) at 2.9 Å was solved by molecular replacement using the structures of each individual component previously solved by our groups ([4] and this manuscript) (Table 1). The asymmetric unit of the crystal showed a complex composed of a single copy of each component (Fig. 5a), confirming the 1:1 stoichiometry previously proposed for the complex between Stl and dimeric Duts[7,10]. To form the BovI-Stl$^{C-ter}$–DutφO11 heterocomplex, both proteins exploit their own homodimerization structural elements in such a way that each protein is reciprocally mimicking the other partner (Supplementary Fig. 9). Stl provides the N-terminal part of its dimerization helices α11 and α13, while DutφO11 contributes with the dimerization helices α2 and α5, generating a surface of interaction of about 1100 Å$^2$ (Fig. 5a, Supplementary Fig. 9). Superimposition of the BovI-Stl$^{C-ter}$ components of the DutφO11–Stl heterocomplex and the BovI-Stl$^{C-ter}$ homodimer shows that the Stl helices α11 and α13 of the second chain in the Stl homodimer occupy similar positions to the DutφO11 helices α2 and α5, confirming structural mimicry as the molecular mechanism used by Stl to interact with dimeric Duts (Fig. 5b). In this way, DutφO11 residues I34, V38, F41, E42, N45 and T49 from helix α2 mediate contacts for both Dut homo and hetero-dimerization (Fig. 5b, Supplementary Fig. 9; Supplementary Tables 1 and 6). Similarly, Stl residues I181, I184, H188, Y191 and F196 in Stl α11, and R253, M242, and Y250 in Stl α13 interact with Dut helix α2 in heterodimerization, while also participating in Stl homodimerization (Fig. 5b, Supplementary Fig. 9;

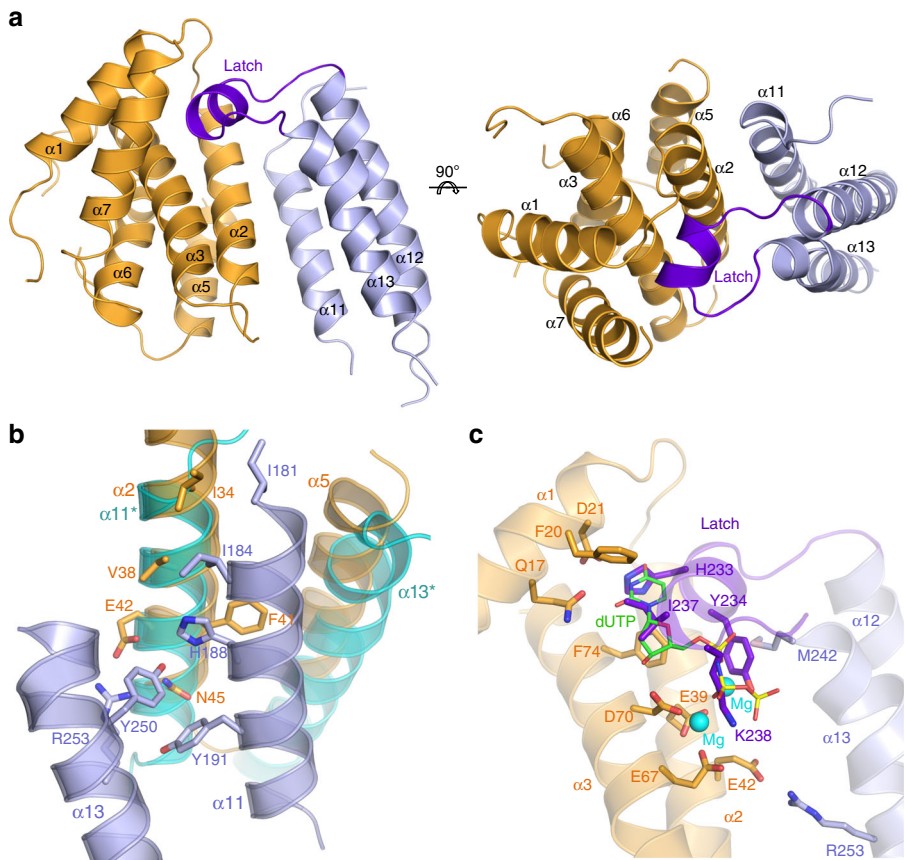

**Fig. 5** Crystal structure of the BovI-Stl$^{C-ter}$–DutφO11 complex. **a** Cartoon representation of BovI-Stl$^{C-ter}$ in complex with the dimeric Dut from phage φO11. DutφO11 is coloured in orange and BovI-Stl$^{C-ter}$ in blue, with the "latch" that projects over the Dut active centre highlighted in purple. Structural elements from both proteins are labelled. Two orthogonal views are shown. **b** Partner mimicry by Stl. Detailed view of the BovI-Stl$^{C-ter}$–DutφO11 heterodimerization helices coloured as **a** superimposed on the BovI-Stl$^{C-ter}$ homodimer showing the dimerization helices from the second BovI-Stl$^{C-ter}$ protomer in cyan. The interacting residues in the BovI-Stl$^{C-ter}$–DutφO11 complex are shown in sticks and coloured by atom type with the carbon atoms in the identical colour of the corresponding molecule. Structural elements are labelled. **c** Stl latch mimics interactions of the dUTP substrate with DutφO11. Detailed view of the BovI-Stl$^{C-ter}$ projecting its latch onto the DutφO11 active centre, coloured as in **a**. DutφO11 residues involved in substrate dUTP-Mg recognition and binding, which also mediate interactions with BovI-Stl$^{C-ter}$, are shown in sticks and labelled, as are their BovI-Stl$^{C-ter}$ counterparts. The substrate dUTP-Mg was placed in the active centre of DutφO11 in complex with BovI-Stl$^{C-ter}$ by superimposing the structure of DutφO11-dUTP (PDB 5MIL[4]) and is shown in sticks with carbon atoms in green. The Mg ions chelated by the dUTP are shown as a cyan spheres. The interacting structural elements of DutφO11 and BovI-Stl$^{C-ter}$ are represented in semi-transparent cartoon. The omit electron density map is shown in Supplementary Fig. 13

Supplementary Tables 1 and 6). Additionally, the Stl projects a long loop that connects Stl helices α12 and α13 into the nucleotide-binding site of DutφO11 (Fig. 5a, c). This loop (residues 230–241), which was unstructured in the free form of BovI-Stl (Supplementary Fig. 9), acquires a single turn helix conformation and resembles the DutφO11 "latch" that comes from one subunit to conform and cover the active centre of the second Dut subunit in the dimeric Dut homodimer (Fig. 5c, Supplementary Fig. 9)[4,7]. However, this BovI-Stl "latch" is inserted into the active centre of DutφO11 and occupies the position of the nucleotide (Fig. 5c). From here, the BovI-Stl latch mimics several of the dUTP interactions with the DutφO11 (Fig. 5c; Supplementary Tables 4 and 6). BovI-Stl residues H233 and I237 mimic the pyrimidine ring interactions, with the former residue interacting with the DutφO11 Q17 and D21, and the second with DutφO11 F20 and F74 (Fig. 5c; Supplementary Tables 4 and 6). The main chain of the I237 and the side chain of K238 from the BovI-Stl latch emulate the interactions of the nucleotide ribose ring contacting with DutφO11 D153 and D70, respectively (Fig. 5c; Supplementary Tables 4 and 6). Finally, the interactions of dUTP Mg-triphosphate moiety, which is recognized mainly by the DutφO11 acid catalytic tetrad comprising E39, E42, E67 and

D70[4], are mimicked by BovI-Stl residues Y234 and K238 from the latch and R253 from helix α13 (Fig. 5c; Supplementary Tables 4 and 6). The mimicry in the dUTP interactions produced by the BovI-Stl latch explains why SaPIbov1 Stl inhibits the dUTPase activity of dimeric Duts and also why the dUTP substrate is a competitive inhibitor of the Stl–dimeric Dut interaction[7]. Taken together, the structure supports the idea that the SaPIbov1 Stl C-terminal domain mimics both functionally the substrate dUTP interactions and structurally the partner subunit in Dut dimer formation.

**Functional characterization of Stl-dimeric Dut recognition.** None of the Stl residues identified as contributing to the Stl–trimeric Dut complex were shown to impact on dimeric Dut interaction with the repressor, implying that a different set of residues is involved in the dimeric Dut–Stl complex. Based on the structure of the dimeric DutφO11 in complex with BovI-Stl$^{C-ter}$ a number of mutations were tested in vivo and in vitro to analyse the importance of different Stl residues in this complex. First, we analysed the role of the dUTP–mimetic latch of Stl (residues 231-NHHYDAIKGK-240) by replacing this loop with the linker

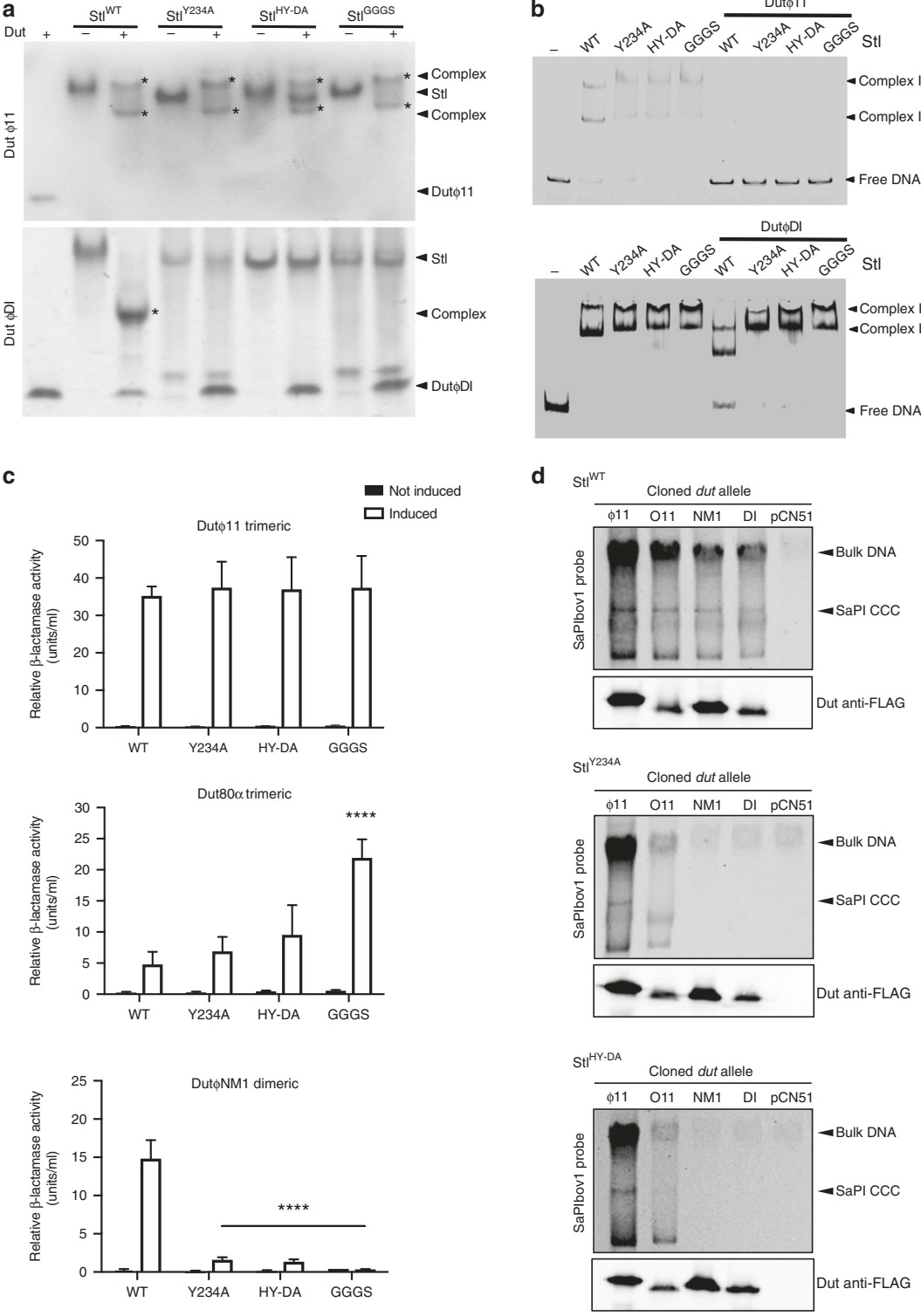

GGGS (Stl^GGGS). In vitro analysis using Native-PAGE showed that this mutation abolished interaction with the dimeric DutφDI, while EMSA assays confirmed that Stl^GGGS maintains the interaction with the target DNA (Fig. 6a, b). As expected for a structural element involved exclusively in the recognition of dimeric Duts, elimination of the latch in the Stl^GGGS mutant has no effect on the interaction with the trimeric Dutφ11 (Fig. 6a). A more refined double-point mutation targeting latch residues H232 (to Asp) and Y234 (to Ala) to generate the Stl^HY-DA mutant was sufficient to abolish interaction with the dimeric DutφDI

while retaining WT DNA binding capacity and WT interaction with the trimeric Dutφ11 (Fig. 6a, b). Furthermore, identical results were seen by the single elimination of the Stl Y234 sidechain (Stl^Y234A) (Fig. 6a, b). The strong effect of this point mutation can be understood due to the fact that Stl Y234 not only mediates contact with the Dut, but also plays a structural role in maintaining the latch architecture by stacking its aromatic ring between the side-chains of the latch residues K238 and M242 (Fig. 5c). We further confirmed the role of the Stl latch in Dut selection in vitro by analysing the capacity of different Duts to

**Fig. 6** In vitro and in vivo evaluation of dimeric Dut recognition and binding by Stl. Mutations $Stl^{Y234A}$, $Stl^{HY-DA}$, and $Stl^{GGGS}$ abolish Stl interaction with the dimeric Duts, but not the trimeric Duts. **a** Native-Page assays show that $Stl^{Y234A}$, $Stl^{HY-DA}$ and $Stl^{GGGS}$ mutants lose their capacity to complex with the dimeric DutϕDI, but retain interaction capacity with trimeric Dutϕ11. **b** Stl DNA-binding capacity of $Stl^{WT}$ and $Stl^{Y234A}$, $Stl^{HY-DA}$, and $Stl^{GGGS}$ mutants is disrupted by trimeric Dutϕ11 as verified by the EMSA assays. Contrarily, the dimeric DutϕDI only disrupts the DNA-binding capacity of $Stl^{WT}$, but not of the mutants, supporting that these Stl variants are incapable of forming complex with dimeric Duts. **c** β-Lactamase assays were performed with strains containing the pJP2085 SaPIbov1 WT or derivatives, and lysogenic for either ϕ11, ϕ80α or ϕNM1 encoding a trimeric or dimeric Dut. Samples were taken after 90 min in the absence or following phage induction with Mitomycin C. All data are the result of three independent experiments. Error bars represent SD. A two-way ANOVA with Tukey's multiple comparisons test compared mean differences between the $Stl^{WT}$ control and the mutants, within columns. Significant adjusted $p$ values were as follows: ϕ80α induced $Stl^{GGGS} \leq 0.0001^{****}$; ϕNM1 induced $Stl^{Y234A} < 0.0001^{****}$, induced $Stl^{HY-DA} < 0.0001^{****}$, induced $Stl^{GGGS} < 0.0001^{****}$, all other values were not significant. **d** SaPIbov1 $Stl^{WT}$ and $Stl^{Y234A}$ or $Stl^{HY-DA}$ island excision and replication following induction of cloned Dut genes. Strains JP6774, JP18043, and JP17679 containing SaPIbov1$^{WT}$, $Stl^{Y234A}$ and $Stl^{HY-DA}$, respectively, were complemented with plasmids expressing either the 3xFLAG-tagged ϕ11 trimeric Dut or ϕO11, ϕNM1 or ϕDI dimeric Duts. Samples were isolated 3 h post-induction with 1 μM $CdCl_2$ and Southern blots were performed using a SaPIbov1 integrase probe. The upper band is "bulk" DNA, including chromosomal, phage, and replicating SaPI. CCC indicates covalently closed circular SaPI DNA. The lower panels below each Southern are western blots probed with antibody to the FLAG-tag carried by the Dut proteins. Source data are provided as a Source Data file

release these Stl mutants from their target DNA. As was observed for the WT forms of Dutϕ11 and BovI-Stl[10,24], the complex formation of trimeric Dutϕ11 with the mutant versions $Stl^{GGGS}$, $Stl^{HY-DA}$ or $Stl^{Y234A}$ disrupts Stl DNA binding capacity and releases the DNA (Fig. 6b). Conversely, the capacity of these mutants to bind the DNA was unaffected by the dimeric DutϕDI (Fig. 6b), in contrast with the WT Stl results and with results previously reported for other dimeric Duts with WT SaPIbov1 Stl[10].

As before, we analysed the impact of these Stl mutations by using the β-lactamase reporter system (Fig. 6c). In correlation with the in vitro results, the $Stl^{GGGS}$, $Stl^{HY-DA}$ and $Stl^{Y234A}$ mutations all abolished the dimeric DutϕNM1 ability to release Stl repression, while the trimeric Dutϕ11 showed a WT phenotype. Again, the different affinity of the trimeric Duts for the SaPIbov1 Stl repressor generated an interesting result: the trimeric Dut80α increased its ability to induce the expression of the reporter repressed by $Stl^{GGGS}$, $Stl^{HY-DA}$ or $Stl^{Y234A}$ (Fig. 6c). Since it is predicted that these Stl mutations reduce the stability of the Stl dimer, the trimeric Dut80α can now de-repress the island more efficiently.

Next, the $Stl^{HY-DA}$ and $Stl^{Y234A}$ mutants were recombined into the SaPIbov1 island to look at SaPIbov1 induction in the presence of the Dut proteins alone on the pCN51 vector, as well as island interference following induction of a Dut-encoding prophage. As expected, Southern blots probing for SaPIbov1 indicated that overexpression of the dimeric Duts DutϕDI, DutϕNM1 and DutϕO11 (from the *Pcad* promoter in expression vector pCN51) in the presence of these Stl mutant islands abolished or reduced the level of island excision and replication compared with the WT SaPIbov1 (Fig. 6d). In contrast, expression of the trimeric Dutϕ11 showed the same levels of SaPIbov1 replication in both WT and mutant SaPIbov1 (Fig. 6d). Again, the western blots show that each Dut expressed similarly in the presence of the different Stls, indicating that the differences seen were due to the Stl mutations (Fig. 6d).

As before, the ability of the different SaPIbov1 mutants ($Stl^{WT}$, $Stl^{HY-DA}$ or $Stl^{Y234A}$) to generate phage interference (ϕ11, ϕ80α or ϕNM1) was examined using phage spot tests. With the WT SaPIbov1, phage infection led to SaPIbov1 induction, causing reduced plaque formation for both dimeric and trimeric Dut encoding phage (Supplementary Fig. 10). However, while phage ϕ11 encoding a trimeric showed reduced plaque formation in the presence of the SaPIbov1 mutants ($Stl^{HY-DA}$ and $Stl^{Y234A}$), the dimeric Dut encoding ϕNM1 did not, confirming that ϕNM1 is unable to induce the SaPIbov1 mutants (Supplementary Fig. 10). Interestingly, this assay confirmed the increased capacity of the trimeric Dut80α to induce the islands carrying the $Stl^{HY-DA}$ or

$Stl^{Y234A}$ mutations. As seen in Supplementary Fig. 10, the ability of the phage 80α to form plaques in the *S. aureus* strains carrying these mutant islands was severely affected since now phage 80α can more efficiently de-repress these mutant islands.

Finally, the in vivo assays, in which we tested induction and transfer of the different SaPIbov1 mutants by the different phages under study, confirmed all the previous results. Thus, while none of the Stl mutations negatively impacted the transfer of the islands by the phages encoding trimeric Duts (ϕ11 and ϕ80α), the Stl mutations significantly reduced the transfer of the islands by the phage ϕNM1, encoding a dimeric Dut (Supplementary Table 5, Supplementary Fig. 8).

**SaPI de-repression is induced by Stl dimer disruption.** Our results suggest that both dimeric and trimeric Duts induce the SaPIbov1 cycle by disrupting the Stl homodimer. In fact, this disruption is required for complex formation with both trimeric and dimeric Duts. The BovI-Stl is reported to be a dimer in solution and we have confirmed this by SEC analysis (Supplementary Fig. 4). However, the BovI-Stl$^{N-ter}$ was monomeric in solution (Supplementary Fig. 4), which is surprising since several 434 Cro repressors and R-M controllers form dimers in solution mainly through the interaction of the HTH and the following helix α5 (ref. [14]). To gain further insight into the relationship between Stl dimerization, SaPI repression and Dut interaction, we analysed our proposed biological model of the SaPIbov1 Stl dimer, which seems to be compatible with Stl box recognition as supported by the in vivo and in vitro results. The model also seems to be compatible with a dimeric organization, since no important clashes are observed and, more importantly, the helix α5 is involved in the dimerization interface, as observed in other members of the HTH_XRE family (Fig. 2a, Supplementary Fig. 3b). In addition, the loop connecting α3–α4 and the beginning of α8 also contributes to the interface, giving a total buried surface in the dimeric model of around 2400.Å$^2$ (calculated by PISA[20]). All the residues provided by helix α5 to this tentative dimerization surface are polar (including the double salt-bridge produced by the interaction of Arg74 and Asp77 (Fig. 7a)), implying that the interaction is mainly hydrophilic. However, PISA analysis gives the dimer a complex formation significance score of 0.0, suggesting that the complex has a low probability of being stable. Given the role played by helix α5 in the dimerization of HTH_XRE family of DNA-binding proteins, we analysed if this element also contributed to Stl dimer stabilization, as our model suggests. Notice that Stl R74 and D77 are also involved in trimeric Dut binding (Fig. 3b, Supplementary Table 3), suggesting that Dut–Stl interaction would directly interfere with Stl dimer formation. We first mutated Arg74, which would participate in a

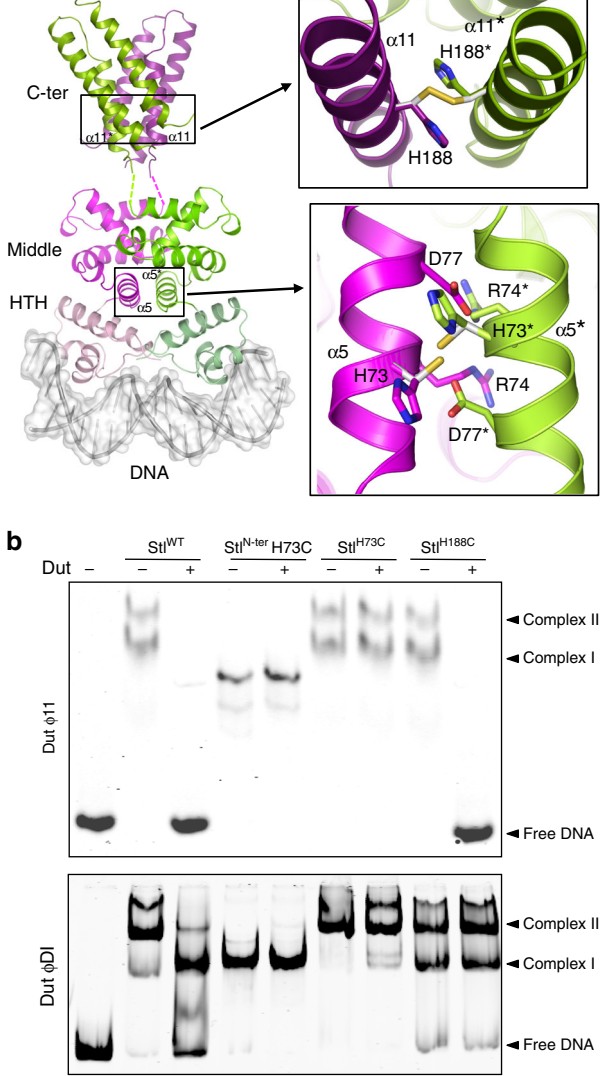

**Fig. 7** Stl dimerization and Dut interaction. **a** Localization of site-specific crosslinks designed based on the dimer model of Stl. (left) A model of dimeric, full-length Stl bound to DNA was generated from BovI-Stl^N-ter and BovI-Stl^C-ter structures using the structure of C.Esp1396I-DNA complex as a template (PDB:3CLC). The Stl model is represented in cartoon, coloured in different tones of magenta for one protomer and green for the other. The tones are light, medium and dark for the HTH, middle and C-terminal domains, respectively. The DNA is represented in surface. The cysteines introduced to induce crosslinking are shown as sticks and coloured, with carbons in white and sulfurs in yellow. **b** EMSA experiments show that the Stl dimers stabilized by H73C or H188C disulfide bonds bind to the SaPIbovI Stl promoter similarly to the wild-type protein. However, these covalent dimers, but not the wild-type protein, are resistant to the DNA dissociation induced by dimeric DutϕDI, while only the Stl dimer mediated by H73C bonds is resistant to the dissociation induced by the trimeric Dutϕ11. Source data are provided as a Source Data file

reciprocal α5 helix salt-bridge, to alanine (Stl^R74A) and this mutant was tested both in vitro and in vivo. Although Stl^R74A presented dimeric behaviour in SEC-MALS (Supplementary Table 2), the thermofluor analysis indicated that the Stl^R74A dimer unfolds around 4° before that of the WT Stl (Supplementary Table 2), supporting the idea that R74 contributes to dimer stabilization. Furthermore, EMSA assays showed that Stl^R74A is highly defective in DNA binding (Fig. 2d) and in vivo

β-lactamase assays confirmed that this mutant had lost the capacity to repress SaPIbov1 (Fig. 2c).

To further confirm our dimerization model, we mutated the helix α5 H73 and the helix α11 H188 to cysteine (Stl^H73C and Stl^H188C, respectively), since our model of BovI-Stl^N-ter dimer and the experimental structure of BovI-Stl^C-ter predicted that these two residues sit at a suitable distance to form a disulfide bond with the equivalent residue in the neighbouring subunit of the dimer (Fig. 7a). SDS-PAGE analysis under non-reducing conditions confirms the formation of a covalent dimer for Stl^H73C and Stl^H188C, the amount of which is increased in the presence of the oxidizing agent cu-phenanthroline (Cu-P) (Supplementary Fig. 11a). When the H73C mutation was introduced in the context of BovI-Stl^N-ter portion alone, a covalent bound dimer can be generated in the presence of Cu-P, unlike the WT BovI-Stl^N-ter that is monomeric in solution (Supplementary Figs. 4 and 11b). Furthermore, EMSA assays showed that the Stl^H73C and Stl^H188C mutants bind DNA and BovI-Stl^N-ter mutated in H73C presents a higher affinity for DNA binding than the WT counterpart (Fig. 7b, Supplementary Fig. 11c). These results support our dimer model and the relevance of the dimeric state for DNA binding. We analysed the Dut-induced inhibition of Stl binding to DNA for the WT, the H73C (full-length and N-terminal portion) and the H188C mutants. EMSA assays showed that unlike Stl WT, the covalent bound Stl H73C mutant, both in its full-length and N-terminal versions, are insensitive to trimeric Dutϕ11 and dimeric DutϕDI and this mutant remains attached to DNA even when high amounts of the Duts are used (Fig. 7b). Notice that wild-type Stl^N-ter is also insensitive to DutϕDI since it lacks the C-terminal domain required to interact with dimeric Duts. In contrast, the trimeric Dutϕ11 but not the dimeric DutϕDI is able to displace Stl^H188C from its target DNA, supporting the greater contribution of the C-terminal domain to the Stl dimer stability (Fig. 7b). Finally, we quantified the binding kinetics of these Stl mutants to trimeric and dimeric Duts by biolayer interferometry (BLI) or microscale-thermophoresis (MST). Both mutations decrease the affinity for dimeric and trimeric Duts, although the effect is greater in Stl^H73C (Table 2). However, introduction of a reducing agent reverses this effect and both mutants recover almost WT affinity for the Duts (Table 2), indicating that this affinity reduction was due to the presence of a certain amount of Stl dimers crosslinked in the sample, as observed in SDS-PAGE (Supplementary Fig. 11a) and correlating Stl monomerization with binding to the target Dut. This fact is confirmed when major crosslinking of the Stl mutants is induced by the presence of the oxidizing agent Cu-P (Supplementary Fig. 11a), which completely abolishes the capacity of binding to both types of Duts (Table 2). The EMSA assays showed that the trimeric Dutϕ11 still conserved some capacity of displace Stl^H188C from the DNA, indicating some Stl^H188C-Dutϕ11 binding although the affinity should be strongly reduced (>1 μM) since we were not be able to measure by BLI.

## Discussion

The life cycle of the PICIs requires an exquisite synchronization with that of their inducing helper phages. To achieve this, the Gram-positive PICIs have developed a unique family of proteins, the Stl (or Rpr) repressors, which sense the entry of the PICI-helper phages to the lytic cycle[3,12]. The SaPIbov1 Stl has been used here as a model of the PICI-encoded Rpr repressors. The remarkable ability of the SaPIbov1 Stl to target unrelated but functionally similar antirepressor proteins relies on the modularity of this repressor. One long-held hypothesis about phage-SaPI evolution has been that phages could evolve to avoid SaPI

**Table 2 Binding affinities of Stl WT and dimerization mutants for trimeric and dimeric Duts**

| | Dutφ11 (BLI)[a] | | | DutφDI (MST)[b] | |
|---|---|---|---|---|---|
| | $K_D$ (M) $(10^{-9})$ | $k_{on}$ $(M^{-1} s^{-1})$ $(10^4)$ | $k_{off}$ $(s^{-1})$ $(10^{-6})$ | $K_D$ (M) $(10^{-10})$ | Signal/noise[c] |
| Stl$^{WT}$ | 1.73 | $2.14 \pm 0.01$ | $37 \pm 6.94$ | 3.03 | 10.5 |
| Stl$^{H73C}$ | 10.3 | $2.45 \pm 0.01$ | $252 \pm 5.29$ | 148 | 48.8 |
| Stl$^{H188C}$ | 2.57 | $2.19 \pm 0.01$ | $56.3 \pm 5.01$ | 63.9 | 15.6 |
| Stl$^{WT}$ + DTT[d] | 6.72 | $4.18 \pm 0.08$ | $281 \pm 22.7$ | 8.60 | 14.8 |
| Stl$^{H73C}$ + DTT[d] | 5.22 | $2.19 \pm 0.02$ | $114 \pm 9.4$ | 12.7 | 12.4 |
| Stl$^{H188C}$ + DTT[d] | 5.10 | $1.54 \pm 0.006$ | $78.4 \pm 4.35$ | 61.3 | 15.1 |
| Stl$^{WT}$ + Cu-P[d] | 11.6 | $2.32 \pm 0.01$ | $270 \pm 5.27$ | 4.24 | 17.1 |
| Stl$^{H73C}$ + Cu-P[d] | NBD[e] | | | NBD[e] | 0.7 |
| Stl$^{H188C}$ + Cu-P[d] | NBD[e] | | | | 3.2 |

[a]Trimeric Dutφ11–Stl interactions were measured by biolayer interferometry (BLI)
[b]Dimeric DutφDI-Stl interactions were measured by Microscale-thermophoresis (MST)
[c]Signal/noise ratio was calculated by dividing the response amplitude by the noise of the run
[d]DTT and Cu-P were used at a final concentration of 1 mM
[e]Not binding detected in experimental condition ($K_D > 1 \times 10^{-6}$ M in BLI or signal/noise <5 in MST)
Source data are provided as a Source Data file

induction by replacement of the gene encoding the phage inducer protein for another that encodes a protein that is structurally unrelated but performs the same function for the phage. Our results call into question the potential for success of such a strategy by supporting the idea that, instead of changing the Stl repressor in response to an anti-repressor substitution by the phage, SaPIs can expand the range of their repressors by recruiting domains able to recognize the new phage-encoded anti-repressors, which incidentally perform the same function for the phage. Targeting a physiological process instead of a unique type of protein is an elegant strategy to increase SaPI transferability, overcoming the phage capacity to escape SaPI induction through the replacement of a specific inducer protein.

In this way, SaPIbov1 Stl presents three domains: one corresponds to an HTH N-terminal domain required to recognize and bind the divergent *stl-str* region present in the SaPI genome; a second (middle) domain responsible for interacting with the trimeric Duts, while the C-terminal part of the protein recognizes the dimeric Duts. The structure shows that the N-terminal and middle domains are closely connected by helix α5, while an unstructured region of about 20–30 residues connects the middle and C-terminal domains. Canonical Cro repressors usually consist of a single dimerization C-terminal domain. Thus, the modular organization observed in SaPIbov1 Stl suggests that the Stl C-terminal domain could have been recruited last, possibly as a consequence of a subset of phages substituting trimeric Duts for dimerics. The absence among SaPIs of an ancestral Stl repressor carrying only one domain can be explained due to the significant disadvantage that its host island would experience in terms of transferability, since most phage would be unable to induce this island by encoding versions of the primitive inducer. Hence, it is unlikely that the less adapted version of the SaPIs, carrying an Stl with a limited ability to interact with the phage inducers, would persist in nature.

Although the modules acquired by the Stl repressor to recognize the target Duts are structurally different, as are the targets themselves, the recognition strategy seems to be similar, in both cases mimicking the interactions mediated by the substrate dUTP, allowing a transitory mechanism of de-repression (Supplementary Discussion). However, differences are observed: for trimeric Duts the Stl middle domain inserts a helix into the active centre, while for the dimeric Duts the Stl C-terminal domain uses a highly flexible loop that acquires a stable conformation only when it occupies the dUTP binding site. Undoubtedly these alternative strategies are due to the wide differences presented by the active centres of the dimeric and trimeric Duts, imposed by

their alternative catalytic mechanisms[28,29]. Additionally, the Stl C-terminal domain also adds a second mimicry mechanism by emulating the dimerization surface of the dimeric Duts. This second mechanism of action must be related to the greater contribution of the C-terminal domain to the dimerization of the Stl repressor, so that the breakage of both dimers and the production of an Stl–Dut heterodimer ensures SaPI derepression,

Site-directed mutagenesis of Stl has allowed us to decipher important residues for DNA and partner recognition. Sequence comparison of SaPIbov1 Stl with homologue Stl repressors from different Staphylococcal species, which are also induced by *S. aureus* phage dimeric and trimeric Duts[4], reveals that the positions proposed for DNA recognition are completely conserved (Supplementary Fig. 12). These observations support the idea that ancestral SaPIs were horizontally transferred not only among *Staphylococcus* species but also to other genus (Supplementary Fig. 12).

Here we have deciphered the fascinating strategy used by the SaPIs to target different families of functionally related phage proteins. This strategy is not confined to SaPIbov1 alone, with the SaPI2 Stl appearing to target multiple families of recombinase[4]. Our results highlight the SaPIs, and more widely the PICIs, as one of the most highly effective and sophisticated phage parasites in nature, giving insight into the mechanism used by these elements to spread through intra- and inter-species transfer.

## Methods

**Bacterial strains and growth conditions.** The bacterial strains used in this study are detailed in Supplementary Table 7. *S. aureus* was grown in Tryptic soy broth (TSB) or on Tryptic soy agar plates. *E. coli* was grown in LB broth or on LB agar plates. Antibiotic selection was used where appropriate.

**Phage spot tests.** SaPI induction occurs upon infection of the host cell by a helper phage encoding the SaPI-inducer protein. SaPIs use a variety of mechanisms to interfere with helper phage replication in order to promote their own packaging and transfer. This detrimental effect on phage reproduction can be utilized to identify SaPI induction by measuring the reduction in phage plaque formation compared with that obtained with the SaPI-free control strain, RN4220. For phage spot tests, the different SaPIbov1 Stl wt and mutant recipient strains were grown to exponential phase and adjusted to 0.35 OD$_{540}$. These were mixed with 8 ml phage top agar (PTA; 2 g/100 ml Nutrient Broth No. 2, Oxoid, plus 0.35 g/100 ml agar, Formedium, and 10 mM CaCl$_2$), and overlaid onto phage base plates (2 g/100 ml Nutrient Broth No. 2, Oxoid, plus 0.7 g/100 ml agar, Formedium, and 10 mM CaCl$_2$). Phage lysates were induced with 2 μg/ml mitomycin C at OD$_{540}$ 0.2 and incubated at 30 ºC, 80 r.p.m. until completely lysed, before filtering with a 0.2 μm syringe filter (Sartorius). Ten microliters of φ11, 80α and φNM1 phage lysates at different dilutions (no dilution, $1 \times 10^{-2}$, $1 \times 10^{-4}$ and $1 \times 10^{-6}$) were dropped onto the different lawn cultures. Three replicates of the phage spot tests were performed with a representative replicate shown.

**DNA methods**. General DNA manipulations were performed using standard procedures. Supplementary Table 8 lists the plasmid constructs used in this study, which were previously described or generated in this work by cloning PCR products obtained with oligonucleotide primers, as catalogued in Supplementary Table 9. Plasmids pETNKI-Stl[N-ter] and pETNKI-Stl[C-ter] for expression of SaPIbov1 Stl deletional variants were produced using plasmid pETNKI-Stl as a template[26]. pETNKI1.10-Stl[N-ter] plasmid-expressing SaPIbov1 Stl residues from 1 to 156 was generated by site-direct mutagénesis introducing a stop codon in pETNKI-Stl using the Stl_Nter_STOP_Fw and Stl_Nter_STOP_Rv primers and Q5-Site Direct Mutagenesis Kit (NEB) (Supplementary Tables 8 and 9). pETNKI-Stl[C-ter] plasmid-expressing Stl residues from 175 to 267 was generated by PCR amplifying the encoding region with the primers FwStl1-8_K175-N267 and RvStlpETNKI1-8 (Supplementary Table 9). The Ligation-Independent Cloning (LIC) system[30] was used to clone the PCR fragment into the pETNKI-his3C-LIC (pETNKI1.1) plasmid (kindly supplied by Patrick Celie, NKI Protein facility) previously digested with KpnI (Fermentas) and treated with T4 DNA polymerase (NEB). Point mutations were generated by Q5-Site Direct Mutagenesis (NEB) using the vector encoding for the WT version of the gene to be mutated as a template or by standard cloning PCR techniques. For protein expression and purification the Stl WT and point mutants were cloned into the pLIC-SGC1-expression vector. All clones were sequenced at IBV Core Sequencing facility or by Eurofins MWG Operon. Detection probes of SaPIbov1 DNA for Southern blots were manufactured by PCR using primers listed in Supplementary Table 9. Probe labelling and DNA hybridization were performed using the protocol for PCR-DIG DNA-labelling and chemiluminescent detection kit (Roche).

**Southern and western blot sample preparation**. Southern and western blots were performed as previously described[3,4]. Duts expressed under control of the inducible Pcad promoter on plasmid pCN51 were used for Southern blots to check SaPI induction following exclusive expression of the Dut protein with SaPIbov1 islands containing either WT or mutant Stl repressors. For strains expressing phage Duts from pCN51, all cultures were grown to OD$_{540}$ 0.2 at 37 °C, 120 r.p.m. in 10 ml TSB supplemented with 10 µg/ml erythromycin and induced with 1 µM CdCl$_2$. Following induction, cultures were incubated for 3 h at 30 °C, 80 r.p.m., and 1 ml samples of each culture were obtained and pelleted.

For examination of island excision and replication in strains containing prophages, all cultures were grown to OD$_{540}$ 0.2 at 37 °C, 120 r.p.m. in 10 ml TSB and induced with 2 µg/ml mitomycin C. Following induction, cultures were incubated for 90 min at 30 °C, 80 r.p.m., and 1 ml samples of each culture were obtained and pelleted.

For Southern blot analyses, samples were re-suspended in 50 µl lysis buffer (47.5 µl TES-Sucrose and 2.5 µl lysostaphin [12.5 µg/ml]) and incubated at 37 °C for 1 h. Fifty-five microlitres of SDS 2% proteinase K buffer (47.25 µl H$_2$O, 5.25 µl SDS 20%, 2.5 µl proteinase K [20 mg/ml]) was added and samples were incubated at 55 °C for 30 min. Samples were vortexed for 1 h with 11 µl of 10× loading dye. Incubation cycles with dry ice and ethanol, then at 65 °C, were performed. Samples were run on 0.7% agarose gel at 25 V overnight. DNA was transferred to a membrane and exposed using a DIG-labelled probe (see DNA methods) and anti-DIG antibody (1:10,000 (v/v), Roche, product 11093274910), before washing and visualization.

Western blots were performed to confirm that the different Duts expressed from pCN51 were expressed at similar levels in each of the different SaPIbov1 Stl mutant backgrounds. Western blot preparation of *S. aureus* samples involved re-suspending pellets in 200 µl digestion/lysis buffer (50 mM Tris-HCl, 20 mM MgCl$_2$, 30% w/v raffinose) plus 1 µl of lysostaphin (12.5 µg/ml), mixed briefly, and incubated at 37 °C for 1 h. 2× Laemmli sample buffer (Bio-Rad, 2-mercaptoethanol added) was added and samples were heated at 95 °C for 10 min, put on ice for 5 min and fast touch centrifuged. Samples were run on 15% polyacrylamide gels and transferred to PVDF transfer membrane (Thermo Scientific, 0.2 µM) using standard methods. Western blot assays used anti-Flag antibody probes (1:2,000 (v/v); Monoclonal ANTI-FLAG M2-Peroxidase (HRP) antibody, Sigma-Aldrich, product A8592) as per the protocol supplied by the manufacturer.

**Protein expression and purification**. SaPIbov1 Stl full-length and its variants (BovI-Stl[N-ter] and BovI-Stl[N-ter] H73C mutant) were expressed from pETNKI 1.10 plasmid (Supplementary Table 8) and purified as described for Stl[26] below. Stl punctual mutants produced from pLIC-SGC1 derivate plasmids and BovI-Stl[C-ter] produced from pETNKI1.1 derivate plasmid were expressed from *E. coli* Rosetta 2 (DE) (Novagen) cultures transformed with the corresponding plasmids (Supplementary Table 8) and purified similarly to SaPIbov1 Stl full-length[26]. Briefly, the culture was grown at 37 °C in LB medium supplemented with the corresponding antibiotics (33 µg/ml chloramphenicol and 33 µg/ml kanamycin or 100 µg/ml ampicillin) up to an OD$_{600}$ of 0.5–0.6 when protein expression was induced with 1 mM isopropyl-b-D thiogalactopyranoside (IPTG) at 20 °C for 16 h. After induction, cells were harvested by centrifugation at 4 °C for 30 min at 3500g, the pellet was resuspended in buffer A (75 mM HEPES pH 7.5, 400 mM NaCl and 5 mM MgCl$_2$) supplemented with 1 mM PMSF and sonicated. The soluble fraction was obtained after centrifugation at 16,000g for 40 min and loaded on a pre-equilibrated His Trap HP column (GE Healthcare). After washing with 10 column volumes of buffer A, the protein was eluted by adding buffer A supplemented with

500 mM imidazole. The eluted protein was digested for His-tag removal using SENP2 (expressed from pETNKI 1.10 plasmid), PreScission (pETNKI 1.1) or TEV (pLIC-SGC1) proteases at a molar ratio 1:50 (protease:eluted protein) for 16 h at 4 °C with slow shaking. After digestion, the sample was loaded one more time into the pre-equilibrated His Trap HP column to remove the His-tag and the protease. The non-retained protein was concentrated and loaded onto a Superdex S200 (GE Healthcare) equilibrated with buffer B (75 mM HEPES pH 7.5, 250 mM NaCl and 5 mM MgCl$_2$) for SEC. The fractions were analysed by SDS-PAGE and those fractions showing purest protein were selected, concentrated and stored at −80 °C. For anomalous X-ray diffraction and phasing BovI-Stl[C-ter] was selenomethionine-labelled (SeMet) by expressing the protein in SelenoMet[TM] Medium Base supplemented with SelenoMet[TM] Nutrient Mix (AthenaES®), according to the manufacturer's indications, and purified as described above. Trimeric and dimeric Duts were expressed from *E. coli* BL21 (DE3) and purified following a similar protocol but using a single buffer consisting on 100 mM Tris pH 8.0 and 150 mM NaCl[4,7,26,27].

**Crystallization and data collection**. The crystals were grown as sitting drops at 21 °C with a vapour-diffusion approach. Initial crystallization trials were set up in the Cristalogenesis service of the IBV-CSIC using commercial screens JBS I, II (JENA Biosciences) and JCSG + (Molecular Dimensions) in 96-well plates. Crystallization drops were generated by mixing equal volumes (0.3 µl) of each protein solution and the corresponding reservoir solution, and were equilibrated against 100 µl reservoir solution. BovI-Stl[Nter] was crystallized at 8 mg/ml in a reservoir solution of 0.2 M Li$_2$SO$_4$, 0.1 M BIS-Tris and 25% PEG3350, this last compound being increased up to 35% to cryoprotect the crystal when freezing in liquid nitrogen. SeMet derivate BovI-Stl[Cter] was crystallized at 19 mg/ml using as reservoir solution 40% PEG3350, 0.1 M BIS-Tris and 0.2 M Na-thiocyanate and was frozen in liquid nitrogen directly from the crystallization drop without the addition of any cryoprotector. Dut–Stl complexes were pre-formed by the mixture of both proteins in a 1 to 1 molar ratio (monomer) in advance of crystallization. BovI-Stl[Cter] in complex with Dutφ011 was crystallized at 20 mg/ml concentration against a reservoir solution of 10% PEG4000, 10% isopropanol and 0.1 M Na-citrate. The cryosolution used for crystal freezing was the reservoir solution increased up to 20% PEG4000 and supplemented with 15% ethylene-glycol. Dutφ11 in complex with BovI-Stl[Nter] was crystallized at 7 mg/ml against a reservoir solution of 24% PEG1500 and 20% glycerol and crystals were frozen directly from the drop. Diffraction data were collected from single crystals at 100 K on ALBA (Barcelona, Spain) and DLS (Didcot, UK) synchrotrons, and processed and reduced with Mosflm[31] and Aimless[32] programs from the CCP4 suite[33]. The data-collection statistics for the best data sets used in structure determination are shown in Table 1.

**Model building**. The structure of the Dutφ11-BovI-Stl[Nter] complex was solved by molecular replacement using Phaser[34] and the structure of the trimeric Dutφ11 (PDB 4GV8[23]) as a model. The initial phases obtained from the molecular replacement were used to generate electron density maps of enough quality to manually build the BovI-Stl[Nter] model in Coot[35]. The BovI-Stl[Nter] structure from the Dutφ11-BovI-Stl[Nter] complex was then used as a model for molecular replacement with the data set collected from the BovI-Stl[Nter] crystals. The structure of BovI-Stl[Cter] was determined by single-wavelength anomalous dispersion (SAD) using a data set from the SeMet derivative BovI-Stl[Cter] crystals. Autosol pipeline of Phenix[36] was used to process the data and to localize one selenium atom, which was enough for calculating experimental phases, and to build the model. The BovI-Stl[Cter] structure and the structure of a protomer of dimeric Dutφ011 (PDB 5MIL[4]) were used as models to obtain the phases by molecular replacement on the data set from the Dutφ011-BovI-Stl[Cter] crystals. All the final models were generated by iterative cycles of refinement using the Phenix wizard[36] and manually optimization with Coot[35]. Data refinement statistics are given in Table 1. Atomic coordinates and structure factors have been deposited in the Protein Data Bank with identification codes 6H48 for BovI-Stl[C-ter], 6H49 for Dutφ011-BovI-Stl[C-ter], 6H4B for BovI-Stl[N-ter] and 6H4C for Dutφ11-BovI-Stl[N-ter].

**Thermofluor**. Thermofluor assays were conducted in the 7500 Real-Time PCR System (Applied Biosystems). Samples of 20 µl containing 5× Sypro Orange (Sigma) and 20 µM of protein in a 75 mM HEPES pH 7.5 and 250 mM NaCl buffer were loaded in 96-well PCR plates. Samples were heated from 20 to 85 °C in steps of 1°. The fluorescence intensity was normalized and analysed using GraphPad Prism software.

**Native gel mobility shift assay**. Purified proteins were mixed at 20 mM 1:1 molar ratio in buffer 75 mM HEPES pH 7.5, 400 mM NaCl and 5 mM MgCl$_2$ and incubated at 4 °C overnight. Samples were loaded into an 8% polyacrylamide gel and electrophoresis was performed at 4 °C. Native gels were stained with coomassie brilliant blue and digitalized with ImageQuant LAS-4000 (GE Healthcare).

**EMSA assays**. Stl DNA binding region for EMSA assays was produced by PCR using Stl_DNA_BR1 and Stl_DNA_BR2 primers modified with 5′ fluorophore IR700 (Eurofins) on *S. aureus* 8325 genomic DNA as a template. The PCR product

includes the SaPIbov1 genome region 13730–13949 which contains the repression site of Stl that controls Str and Xis expression[37]. Purified PCR product (10 ng/µl) and Stl protein were mixed in EMSA buffer (50 mM HEPES pH 7.5, 5 mM MgCl$_2$, 75 mM NaCl and 0.5 mM EDTA). The samples were incubated for 15 min at room temperature. 8% polyacrylamide gels were electrophoresed in Tris-Borate-EDTA (TBE) buffer at 100 V for 1 h, followed by loading of the samples. Electrophoresis was then performed in TBE buffer for about 150 min at 100 V at 4 °C. Gels were analysed by Odyssey Imaging System (LI-COR).

**SEC and SEC-MALS.** SEC analyses were carried out using a Superdex 200 Increase 10/300 GL column connected to an AKTA Pure system (GE Heatlhcare) and equilibrated with buffer 75 mM HEPES pH 7.5, 250 mM NaCl and 5 mM MgCl$_2$. Samples containing 200 µg of protein were loaded into the column and were eluted isocratically at a flow rate of 1 mg/ml. Peaks were collected and checked by SDS-PAGE. Chromatograms were exported and analysed in GraphPad Prism software. In the SEC with multi-angle light scattering (SEC-MALS) experiments the chromatographic system was coupled to a Wyatt DAWN HELEOS-II MALS instrument and a Wyatt Optilab rEX differential refractometer (Wyatt). The Astra 7.1.2 software from the manufacturer was used for acquisition and analysis of the data.

**Di-sulfide bond formation.** Copper-phenanthroline (Cu-P) was used as an oxidizing agent to induce di-sulfide bond formation. Cu-P was prepared by mixing 60 mM CuSO$_4$, 200 mM o-phenanthroline and 50 mM NaH$_2$PO$_4$[38]. Proteins were mixed with Cu-P (1 mM final concentration) and incubated for 5 min at 37 °C and the formation of di-sulfide bonds was checked by SDS-PAGE with no reducing agent.

**β-Lactamase assays.** For the β-Lactamase assays, cells were obtained at 0.2 OD$_{540}$ and at 90 min following mitomycin C phage induction (2 µg/ml) or without induction. One millilitre samples of each culture were obtained and bacterial growth was immediately arrested by addition of 10 mM sodium azide (final concentration) and snap freezing on dry ice. OD$_{540}$ was measured for all samples as a reference for bacterial cell density. β-Lactamase assays, using nitrocefin as a substrate, were performed using an ELx808 microplate reader (BioTek). Fifty microlitres of each culture were defrosted on ice, and then diluted 1:2 (v/v) in 50 mM KPO$_4$ buffer (pH 5.9). Measurement of absorbance at OD$_{490}$ was started immediately following addition of 50 µl nitrocefin working stock (6 µl of nitrocefin stock [23.8 mg/ml anhydrous nitrocefin in DMSO] diluted in 10 ml 50 mM KPO$_4$ buffer, pH 5.9). Plates were read every 20 s for 30 min. Relative β-lactamase activity (units/ml) was defined as (slope)[1/(A540)(d)(V)], where slope is the Δabsorbance/hour, $A_{540}$ is the absorbance of the sample at OD$_{540}$, $d$ is the dilution factor, and $V$ is the sample volume.

**Biolayer interferometry.** The kinetics parameters of the interaction, binding affinity ($K_D$) and rate constants of association ($k_{on}$) and dissociation ($k_{off}$), between trimeric Dutφ11 and Stl WT and Cys mutants were measured by biolayer interferometry (BLI) using the BLITz system (FortéBio). Proteins were diluted in Stl buffer (75 mM HEPES pH 7.5, 250 mM NaCl and 5 mM MgCl$_2$) and the assays were carried out in the same buffer. When necessary the buffer was supplemented with the corresponding DTT or Cu-P at 1 mM final concentration and the samples were incubated for 10 min at 37 °C. For each interaction, the His-tagged Dutφ11 was immobilized on Ni-NTA biosensors (ForteBio) at 1 µM concentration. At least four different dilutions of Stl (from 1 to 0.0125 µM plus the reference without Stl) were used in the association and dissociation steps for each Stl:Dut interaction measured, adjusting the highest concentration of Stl to 10 times the estimated $K_D$. Kinetics values calculation and data analysis were performed with BLITz Pro 1.2 software. A 1:1 model was employed to fit the data.

**Microscale thermophoresis.** His-tagged Stl WT and Cys mutants were fluorescently labelled with NT-647 amine reactive dye from the Monolith NT Protein Labeling kit RED-NHS (NanoTemper Technologies) according to the manufacturer's instructions. Briefly, protein was mixed with 3× dye solution, incubated 30 min in dark and then purified by gravity column with buffer A to remove free dye. Labelled Stl protein was used at a final optimized concentration of 150–300 nM (depending on the affinity for the Dut). For the measurements of direct purified proteins both Stl and Dut were directly prepared in buffer A (75 mM HEPES pH 7.5, 400 mM NaCl and 5 mM MgCl$_2$). For measurements in the presence of DTT, the Stl protein was prepared in buffer A supplemented with 2 mM DTT (final concentration 1 mM when diluted in Dut solution). A 16-point twofold dilution series (ranged from 10 to 0.3 µM) of Dut in buffer A was mixed with Stl-labelled solution (1:1). After 15-min incubation at room temperature, samples were filled into Premium Coated Capillaries (K005, NanoTemper Technologies) and MST measurements were performed on a Monolith NT.115 (NanoTemper Technologies) in RED channel using 20% of LED excitation power and 40% of MST power. Results analyses were performed with M.O. Affinity Analysis software (NanoTermper Technologies) as described[39].

**Statistical analyses.** As indicated in the figure legends, two-way ANOVA comparisons with either Tukey's or Sidak's multiple comparisons tests were conducted or one-way ANOVA comparisons, as appropriate. All analysis was done using Graphpad Prism 6 software.

**Protein assemblies and interactions analysis.** PISA server[20] and PDBsum[40] were used to explore macromolecular interfaces and predict probable biological assemblies. Analysis of BovI-Stl$^{Nter}$ structure did not reveal any specific interaction that could result in the formation of stable quaternary structure. Analysis of BovI-Stl$^{Cter}$ with PISA server proposed a dimeric organization with the highest Complexation Significance Score (CSS = 1). BovI-Stl$^{Cter}$ dimerization is generated by exploiting a twofold crystallographic axis and produces a gain in the free energy of solvation of −6.5 kcal/mol (per subunit) with a $P$ value of the solvation energy gain as low as 0.088. These values imply that the proposed assembly should be biological relevant. The interactions described in the manuscript are the consensus result of the analysis of the corresponding structures with PISA server[20], PDBsum[40] and Contact[33] software.

**Structural modelling of Stl–DNA complex.** The structural model of BovI-Stl$^{Nter}$ in complex with its target DNA box was generated by superimposing the HTH domain of BovI-Stl$^{N-ter}$ (residues 12–83) with the equivalent portion (residues 7–72) of the controller protein from the Esp1396I restriction-modification system (C.Esp1396I) in complex with its DNA box (PDB 3CLC)[41]. To generate the dimeric Stl, a second BovI-Stl$^{N-ter}$ monomer was similarly superimposed in the second subunit of the dimeric C.Esp1396I repressor. The DNA–Stl model was finished by substituting the DNA sequence recognized by the C.Esp1396I repressor by the 17-mer TATCT-CaatttGAGATA sequence (invert repeats in capital letters) corresponding to the SaPIbov1 Stl binding site[21]. In the DNA sequence substitution, the position of the invert repeats recognized by both repressors was matched. Energy minimization of the model was then performed in parallel by: (i) the minimize structure module of Chimera[42] with 100 initial Steepest descent steps followed of 100 conjugated gradient steps, and (ii) Yasara energy minimization server[43] using the default Yasara force field and minimization values. Both software programmes generate almost identical models (RMSD of 0.4 Å for the Cα atoms superposition of both dimeric Stl models) with similar DNA recognition (analysed with DNAproDB[44]). Analysis with Contact[33] showed that no clashes between subunits in the dimer occur (shortest distance between side-chains was 2.65 Å). The model generated with Chimera was used to produce the figures presented in the manuscript.

**Reporting summary.** Further information on research design is available in the Nature Research Reporting Summary linked to this article.

## Data availability

Coordinates and structure factors have been deposited in the Protein Data Bank under accession code 6H48, 6H4B, 6H49, 6H4C. Source Data underlying Figs. 2c-d, 4a-c, 6a-d, 7b, as well Supplementary Figures 2, 4, 7, 8, 10 and 11a-c and Table 2 are provided as a Source Data file. The authors declare that all other relevant data supporting the findings of this study are included in this published article and its Supplementary Information files, or from the corresponding authors upon request.

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

## Acknowledgements

We would like to thank The IBV-CSIC Crystallogenesis Facility for protein crystallization screenings. The X-ray diffraction data reported in this work was collected in experiments performed at XALOC and I.04 beamlines at ALBA and DLS Synchrotrons, respectively. Preliminary and complementary X-ray diffraction experiments were performed in I.03 and I.24 beamlines at DLS synchrotron. We thank the staff of the beamlines used at these synchrotrons for assistance in the measurement of the crystals. This work was supported by grant BIO2016-78571-P from the Ministerio de Economia y Competitividad (Spain) and grant Prometeo II/2014/029 from Valencian Government (Spain) to A.M., and grants MR/M003876/1 and MR/S00940X/1 from the Medical Research Council (UK), BB/N002873/1 and BB/S003835/1 from the Biotechnology and Biological Sciences Research Council (BBSRC, UK), Wellcome Trust 201531/Z/16/Z, and ERC-ADG-2014 Proposal no. 670932 Dut-signal from EU to J.R.P. C.A. and J.R.C were supported by FPI BES-2014-068617 and FPU13/02880 predoctoral fellowships respectively. X-ray diffraction data collection was supported by Diamond Light Source block allocation group (BAG) Proposal MX14739 and MX16258 and Spanish Synchrotron Radiation Facility ALBA Proposal 2016071762 and 2017072262. J.R.P. is thankful to the Royal Society and the Wolfson Foundation for providing him support through a Royal Society Wolfson Fellowship.

## Author contributions

Conceptualization: A.M. and J.R.P.; methodology: A.M. and J.R.P; investigation: J.R.C.-T., C.A., S.H., J.D., J.B., X.S., A.M. and J.R.P; writing—original draft, A.M. and J.R.P; funding acquisition: A.M. and J.R.P.; resources: A.M., J.R.P., J.R.C.-T., C.A., S.H., J.B., J.D. and X.S.; supervision: A.M. and J.R.P.

## Additional information

**Competing interests:** The authors declare no competing interests.

