## [Peer Review File · Nature Communications]

Reviewers' comments:

Reviewer #1 (Remarks to the Author):

The manuscript by Ciges-Tomas et al reports the structure of the SaPI repressor StI and explores the mechanistic basis for StI interactions with phage encoded dUTPases. This is the first solved structure of this important protein and helps to elucidate how SaPIs parasitize their cognate helper phage at the molecular level. There are several points raised below that need to be addressed prior to acceptance: importantly, the manuscript is missing key controls and analyses, and at times the writing is so unfocused/unclear, that it prevents the reader from understanding the findings.

Major points

The overall text needs trimming and the point of the paper needs to be clarified: The introduction is focused on the author's work in deciphering the molecular mechanism of StI function and regulation, but the discussion focuses on the evolutionary strategy, however none of the results in this paper are appropriate for addressing evolution. Therefore, the title of the paper is not appropriate and does not reflect the findings. This is not to say the findings are not impressive and interesting, however, the authors are way over reaching on how their results are related to evolution. A central argument here is outlined in Line 535 – re: recruiting domains, this is far-fetched, are there StI proteins without the additional domain? Such comparative approaches would be needed to even start to make this argument, however, that is not a central finding of the data presented.

A major finding revealed by this paper is that StI targets the functionality of the dUTPases (regardless of trimeric/dimeric), StI hits all the conserved residues and the authors do not make clear what is conserved and not conserved between the trimeric and dimeric Duts. This information should be added (ie conserved catalytic motifs are brought up in line 279, but the average reader does not know these motifs from previous work and so this should be included in the supplement). I am also puzzled by the conclusions as StI is NOT targeting totally unrelated proteins (it targets the same active site in the Duts) so conclusions along these lines are overstated.

In the introduction, a section on Duts and their relevance to the phage lifecycle is missing. The authors focus a significant amount of the results section on the molecular mechanism of StI inhibition on Duts, therefore background information on what is known about Dut function/inhibition by StI is needed. Is StI inhibition of Dut activity important for SaPIs to inhibit/reduce phage fitness? it would be helpful to discuss the biological context for inhibiting Duts, when are Duts expressed, and the necessity of Duts for phage viability.

Specific changes needed in the results/discussion:

Figure 1A – Needs additional labels marking the section of the protein that corresponds to each crystal structure solved and labeled with the name used for this construct

Overall there was too much extraneous information in the first part of the results section. Lines 114-135 - the preamble of the explanation as to why the authors didn't solve the full length structure should be eliminated. Instead the authors just say: Due to the low complexity linker we solved 2 constructs of the protein (X and Y)

Line 117 – states "C-terminal portion of unknown function", yet in line 126-128 explains that C-term recognizes dimeric Duts with same reference 5. Pertinent background information should be in the introduction.

Lines 132 and FigS1A: S1A is incomplete: The N-term needs to also be tested for interaction with the trimeric Dut, and controls are needed showing , trimeric not interacting with C, Dimer not with N. Figure 4 shows full length StI function, but not just N-term.

Line 156 – Explanation of how PISA software is used to identify proposed dimeric interface would strengthen the analysis, also there must be values or parameters for this, and the use of the software need to be addressed in the methods.

Line 166: spell out RMSDs, and how is this deviation is significant or not significant?

Line 170 - Cro is brought up for first time, would be helpful to have some of this information in the intro, especially as it is a recurring theme (as it is brought up again in results/discussion several times (lines 582, 565, 546, 463)

Line 174 - For the model – The authors need a methods section explaining how they generated this model. Also energy minimization should be performed on this model before any structural analysis. The authors analysis of this model includes a discussion on specific interactions of Stl with its DNA recognition sequence (Stl box sequence/references not shown, referred to again in line 467), but the authors do not explain and illustrate atom to atom interaction between DNA and protein. If they were to keep this analysis, it would be necessary to model in the exact recognition sequence of Stl in order to identify if the electrostatic interactions in 2B make sense; they should highlight non-specific vs specific interactions in this analysis and a figure highlighting the location of the Stl box recognition sequence in respect to the dimer on their model to illustrate that the dimer “length” matches the length of the recognition sequence. Thus as it stands the conclusion in Line 468 is invalid. The Stl Box sequence and references are needed, and it is not sufficient to just say that the dimer “seems to be compatible with Stl box recognition”.

All references to “Indirect” interactions with DNA should be changed to “nonspecific” (those interactions with DNA backbone) for example, Line 189 “Indirect read-out”

For Fig 2A - completely remove A, the Authors are not focusing on the dimer and this is confusing given they solved it as a monomer (and the dimerization comes up later).

Fig 2B could be in the supplement with Figure S2. This figure allows the authors to say that the helix A3 is close to DNA and could be interacting with the major groove, and to test those residues in panels C and D. However, for these point mutations, the Authors are lacking any negative controls (ie residues that are not predicted to interact with the DNA).

Lines 213-216- the Authors state that they confirmed the mutants were dimers in solution and that they had the same denaturalization TM range as WT protein but they only show 2 mutants, and the mutants do not match WT in SEC. The SEC needs to be shown for every mutant, and since the S44A is of concern - they should only be testing mutants that better recapitulate the WT protein, more like the R51A.

Figure 2C – has extraneous information and missing controls. The beta-gal assays in the presence of Duts are irrelevant, the point is the EMSAs show DNA binding is inhibited, and uninduced beta-gal assays WT vs mutant are consistent with loss of repression. However, to show that these proteins are indeed functional they should be showing interaction with Duts via native page for example. This would confirm that their proteins are not just misfolding and therefore not binding DNA

Line 229 - ‘complex formation induces reduced changes’ not clear what reduced changes are.

Mention in line 225 solved by molecular replacement (using what? Include PDB ID) and they didn’t mention how the N-terminal and C-terminal structures were solved (in Figure 1).

Figure 3A – The color scheme can be refocused for better clarity- such as monochrome for the Duts, and keep the color scheme of Stl from the previous figure (HTH/Middle etc), there needs to be a focus on how Dut binding leads to inhibition of Stl binding to DNA, they focus more on the function of Dut and how Stl mitigates Dut activity. Hence again why the introduction needs to expand more upon how Dut activity is blocked by Stl. Also for this figure, where is the Dut-Stl interface relevant to the function of Stl? Where is the Stl DNA binding interface in relation to the Stl-Dut interface?

Figure 3B/C: The motifs need to be labelled more clearly (with different colors) as I can't tell which are which. It would be helpful to have a supplementary figure of the trimeric Dut with the Motifs labeled (Motifs referred to in line 245)

Figure 3C and line 254 - they don't actually show mimicry, this looks like it is actually just occluding the nucleotide binding site, the tyrosine is not overlaid perfectly so is not mimicry, all wording/claims of mimicry here and similar claims elsewhere should be removed (line 276; in reference to Figure S7 the authors again discuss mimicry, however, the data presented does not prove mimicry (do the residues match, or do the residues actually interact? Would need to see hydrogen bonds or van der waals interactions).

Figure 4A - labelled Y112113A should be re-labelled StIYYAA. The take home for this should be simplified, the important finding is that the double mutation doesn't respond to trimeric Duts, but doesn't stop the response to Dimeric, don't show the point mutations that didn't work here. In general the authors need to simplify the data that they are showing to make it accessible to readers.

Lines 319-324 Based on the southern blot data shown, it not convincing that there is absolute selectivity of the trimeric vs dimeric, ie the dimeric D1 Dut is not inducing the StIYYAA. In the last western for StIYYAA- the NM1 Dut is running lower than it did in previous panels, these should also be normalized in some way to reflect uneven expression of Duts.

In figure S6, the authors show that the StIY112A is still inhibitory to phi11 with trimeric DUT, even though in Figure 4b, they show that this point mutation results in no SaPI excision/replication with expression of phi11 Dut. By spot assay, the YYAA and Y112A should be identical, and they are not? Further, the spot assay in Figure S6 should be done with D1, since they show that it still interacts by Native gel in 4C, but in 4B southern blot looks like the SaPI is not de-repressed.

Figure 5 - line 367 - since referencing the surface, it would be helpful to show a surface view of each protein, with the surface highlighted.

Lines 361-400 - discussion here is way too long and does not actually reflect what is being shown in the figures. For example, In lines 376-380 - no contacts are actually shown the figure, they are just listing residues that appear in proximity, but there is no indication of what the rubric is for deciding an interaction. It appears as though residues are near each other and the authors conclude mimicry, this is not justified.

If the latch is mimicking the Dut dimers own latch, is the Dut active as a dimer? How then does the latch differ from the Dut latch (as it must, or Dut wouldn't be active), again suggests not mimicry. In line 384 - they say that it occupies the position of the nucleotide but then say mimicking several of its interactions, but this isn't shown: in S8 indeed H233 doesn't even look like it is in the same position. Rather than mimicking it looks much more like competitive binding. The discussion of the magnesium in S8 in line 393, there isn't a tetrad in the figure so how are we supposed to see mimicry?

Figure 6A the interaction between trimeric Dut - there is a change in banding pattern to Figure 4C (where 2 bands form indicating the interaction, 6A with WT StI is not convincing despite the asterisk), is this a reliable technique? How many times were the assays performed? Are replicates going to be shown in the supplement?

Line 413 - why would you expect only a minor effect, shouldn't it have no effect? The gel in Fig 6B is made of 2 different gels that are pieced together, since the control for the gel is on the far left (WT without Dut) this should be repeated. Looks sort of like the C-term mutations make binding to the trimeric Dut tighter, and therefore more free DNA, is this true?

Line 421 - referring to aromatic ring, this should be a separate figure.

Line 469 - what is an important clash, need to show clash analysis (software available).

Line 470 - The authors state that dimerization interface is provided by helix 5, however the N term segment was a monomer as revealed by their SEC and structures, so how can this be concluded?

Cannot do the analysis in Figure 7A without first minimizing the model as mentioned above also show the hydrogen bonds and any salt bridges as dashes on the structure.

The band in the EMSA in figure 7B StIH188C without Dut looks off, perhaps that mutant is defective in DNA binding and it shouldn't be included in the analysis.

Minor points:

Legend for Fig 4 -B and C are mixed up.

Lines 241-244 can be removed, not informative.

Line 410 - missing a word between that maintains (should be that StIGGGS maintains the interaction)

Typos:

Line 589 - has be able to

592 - side-directed

Line 529 - replace hypothetical belief with hypothesis

Line 379, R235-→R253.

Reviewer #2 (Remarks to the Author):

The manuscript by Marine and colleagues described the means by which phage proteins can directly bind to and de-repress a master regulator in *S. aureus* in order to propagate the content of a pathogenicity island. Prior studies from this group showed that binding of sequence and structure divergent inducer protein proteins to the a master repressor results in island induction. Here, the investigators have used a structure-based approach to analyze the details of how divergent proteins (specifically dUTPases) can be enraged by the same repressor (bov1) to relieve repression. The work is nicely done but the style of presentation is really difficult. Constant use of jargon, and insufficient background details in both the Abstract and Introduction thwart what is an otherwise very elegant study. The authors are advised to consider that the readership of Nature Comm. extends beyond the microbiological community, so a manuscript that is more tailored to a broader scientific audience should be the goal.

A second major concern is that much of the work is focused on how depression is mediated by disrupting dimer formation. However, no quantitative details for any of the binding proteins is provided. Even a simple analysis using quantitative native gels would provide a much stronger supporter for their hypothesis.

If the authors are able to address each of these concerns, this manuscript will be sufficiently above the high bar for a journal such as Nature Comm.

Minor points:

1. Figures 3C and %c should show an omit map around the residue that mimic interactions at the dUTP-binding pocket.

2. Table 1 should include the Molprobit clash score. Also, what is "others" in 3rd line above Ramachadran refer to? There should be some clarification here.

Reviewer #1

The manuscript by Ciges-Tomas et al reports the structure of the SaPI repressor StI and explores the mechanistic basis for StI interactions with phage encoded dUTPases. This is the first solved structure of this important protein and helps to elucidate how SaPIs parasitize their cognate helper phage at the molecular level. There are several points raised below that need to be addressed prior to acceptance: importantly, the manuscript is missing key controls and analyses, and at times the writing is so unfocused/unclear, that it prevents the reader from understanding the findings.

Major points

The overall text needs trimming and the point of the paper needs to be clarified: The introduction is focused on the author's work in deciphering the molecular mechanism of StI function and regulation, but the discussion focuses on the evolutionary strategy, however none of the results in this paper are appropriate for addressing evolution. Therefore, the title of the paper is not appropriate and does not reflect the findings. This is not to say the findings are not impressive and interesting, however, the authors are way over reaching on how their results are related to evolution.

We are really sorry for our incapacity to show in the original manuscript the link between StI structure and SaPI evolution. One of the most exciting things about the StI repressor is its ability to interact with unrelated proteins. This ability is absolutely essential for the spread of the SaPIs in nature. Regarding the case analysed here - the SaPI_{bov1} StI repressor - it is unlikely that this protein evolved *de novo* with the ability to interact with both dimeric and trimeric Duts. Thus, it is logical to assume that the StI interacted initially with one of these proteins. Since SaPI induction is detrimental for the phage (the phage titre can be reduced more than 1000 times in presence of the induced SaPI), we have previously demonstrated (PLoS Genet. 2015; 11(10):e1005609) that this negative effect selects for viruses that have modified/changed the SaPI inducer. Since phages have mosaicism, the easiest way to do this is to replace the gene encoding the inducing Dut with another one encoding a different Dut type. Herein lies the evolutionary link: in this scenario, and to persist in nature, the StI must evolve to acquire the ability to interact with both Dut types. Otherwise, it will disappear. Thus, it is very clear that there is an evolutionary driving force that "has forced" the SaPI_{bov1} StI repressor to acquire its current structure. In this paper we have deciphered the structure that has evolved to enable the SaPI_{bov1} StI to persist in nature in response to the evolutionary challenge imposed by the phages when they modified their original inducer.

We have now modified the introduction to show this link clearly.

Following the indications of the reviewer we have modified the title of the manuscript and now reads: "A polyamorous repressor: deciphering structurally the evolutionary strategy used by phage-inducible chromosomal islands to spread in nature".

A central argument here is outlined in Line 535 – re: recruiting domains, this is far-fetched, are there StI proteins without the additional domain? Such comparative approaches would be needed to even start to make this argument, however, that is not a central finding of the data presented.

As indicated previously, the acquisition of an extra domain, allowing the StI to interact with a novel family of inducers, would provide a huge advantage for the SaPI in terms of transfer. In this new scenario, it is unlikely that the less well-adapted version of the SaPIs, carrying an StI with a limited ability to interact with phage inducers, will persist in nature.

We have now introduced the following paragraph in the Discussion to reflect this:

“If our hypothesis is true, and the StI repressors have evolved by recruiting additional domains, why we did not find an ancestral StI just carrying one domain? It is obvious that the island carrying this repressor would have a huge disadvantage compared to that with multiple domains in terms of its transferability, since most phage would be unable to induce this island by encoding versions of the primitive inducer. In this new scenario, it is unlikely that the less adapted version of the SaPIs, carrying an StI with a limited ability to interact with the phage inducers, would persist in nature. ”

A major finding revealed by this paper is that StI targets the functionality of the dUTPases (regardless of trimeric/dimeric), StI hits all the conserved residues and the authors do not make clear what is conserved and not conserved between the trimeric and dimeric Duts. This information should be added (ie conserved catalytic motifs are brought up in line 279, but the average reader does not know these motifs from previous work and so this should be included in the supplement). I am also puzzled by the conclusions as StI is NOT targeting totally unrelated proteins (it targets the same active site in the Duts) so conclusions along these lines are overstated.

To answer the interesting questions raised by the reviewer, we have now included this paragraph in the introduction:

“Of note is the fact that trimeric Duts are all- β proteins with three independent active sites formed by the contribution of five conserved motifs (named I-V), with each subunit participating in the formation of each active site present in the trimer¹⁰ (Supplementary Fig. 1). Conversely, the dimeric Duts are all- α proteins and their active centers are also generated by five conserved motifs, although the sequences of these motifs are totally different from those on trimeric DUTs^{8,9} (Supplementary Fig. 1). These differences are reflected in the alternative architectures of the corresponding active centers (Supplementary Fig. 1) and, consequently, different catalytic mechanisms for each family of DUTs^{8,17,18}. ”

We hope that now it will be clear that trimeric and dimeric are *structurally* unrelated proteins and that their active centers are completely different, although both types of proteins catalyze the hydrolysis of the same substrate. The fact that the number of motifs conserved in both types of enzymes is the same (five) can lead to the confusion that these are the same in dimers and trimers, which is not the case. Following the indications of the reviewer, and to clarify this point, we have added an additional Figure (Supplementary Fig. 1) highlighting these motifs in the structures and sequences of both dimeric and trimeric dUTPases.

In the introduction, a section on Duts and their relevance to the phage lifecycle is missing. The authors focus a significant amount of the results section on the molecular mechanism of StI inhibition on Duts, therefore background information on what is known about Dut function/inhibition by StI is needed. Is StI inhibition of Dut activity important for SaPIs to inhibit/reduce phage fitness? it would be helpful to discuss the biological context for inhibiting Duts, when are Duts expressed, and the necessity of Duts for phage viability.

It is a mystery what is the function of the Dut proteins in the phage life cycle. As previously reported by our groups, deletion of the *dut* gene does not impact the phage cycle under laboratory conditions. However, the fact that all *S. aureus* phages encode either a trimeric or a dimeric Dut indicates that this protein is relevant in nature. This information has been now included in new version of the manuscript: *“Interestingly, the function of these proteins in the phage life cycle remains a mystery. As previously reported by our groups, deletion of the phage dut genes does not impact the phage cycle under laboratory conditions^{4,6}. However, the fact that all S. aureus phages encode either a trimeric or a dimeric Duts indicates a relevant role for this protein in nature.”*

Specific changes needed in the results/discussion:

Figure 1A – Needs additional labels marking the section of the protein that corresponds the each crystal structure solved and labeled with the name used for this construct.

Following the reviewer’s comments the sequence has divided in the two complementary portions corresponding to the crystal structures solved and has been labeled with the names used for the corresponding fragments in the manuscript.

Overall there was too much extraneous information in the first part of the results section. Lines 114-135 - the preamble of the explanation as to why the authors didn’t solve the full length structure should be eliminated. Instead the authors just say: Due to the low complexity linker we solved 2 constructs of the protein (X and Y).

Following the reviewer’s suggestion, this section has been significantly shortened.

Line 117 – states “C-terminal portion of unknown function”, yet in line 126-128 explains that C-term recognizes dimeric Duts with same reference 5. Pertinent background information should be in the introduction.

We have now included the following information in the introduction: *“What evidence exists to support the hypothesis that SaPI Stl proteins have multiple domains to interact with unrelated proteins? Although no Stl repressor structure has been solved, previous studies with the SaPIbov1 Stl repressor proposed an architecture with an N-terminal DNA binding domain and a C-terminal portion of unknown function that seems to consist of two domains connected by a low complexity segment^{5,11}. We demonstrated that the deletion of the C-terminal portion after the low complexity segment generates a SaPIbov1 Stl that retains the capacity to interact with trimeric but not dimeric Duts, while the deletion of the N-terminal DNA binding domain has the opposite effect, precluding Stl binding to trimeric but not to dimeric Duts⁸. These results suggested that Stl is composed of different domains with alternative and complementary functional characteristics”.*

Lines 132 and FigS1A: S1A is incomplete: The N-term needs to also be tested for interaction with the trimeric Dut, and controls are needed showing, trimeric not interacting with C, Dimer not with N. Figure 4 shows full length Stl function, but not just N-term.

We previously showed (Bowring et al, Elife, 2017) that the Stl N-terminal portion interacts selectively with trimeric Duts while the C-terminal portion, longer than that the described here (residues 87-267), interacts selectively with dimeric DUTs. In this manuscript, we use a shorter version of Stl (residues 175-267) and now in the new Supplementary Figure 2 we confirm that this small fragment conserves the capacity to interact selectively with dimeric DUTs. In the

new Supplementary Figure 2, as the reviewer suggested, we have replicated the assays with a trimeric Dut, confirming that these Duts do not interact with the C-terminal portion of Stl.

Line 156 – Explanation of how PISA software is used to identify proposed dimeric interface would strengthen the analysis, also there must be values or parameters for this, and the use of the software need to be addressed in the methods.

We have added a Methods section describing the use of the software and the values (and significance) of the PISA analyses. The result of this analysis, which supports the interaction and oligomerization conclusions, is also reported in the manuscript.

Line 166: spell out RMSDs, and how is this deviation is significant or not significant?

RMSD has been spelled. The values of RMSD support that the structures present similar folds.

Line 170 - Cro is brought up for first time, would be helpful to have some of this information in the intro, especially as it is a recurring theme (as it is brought up again in results/discussion several times (lines 582, 565, 546, 463)).

Some information about the *ci*/Cro family of phage repressor has been now included in the introduction section. The new paragraph in the introduction reads: *“Functionally, the Rpr proteins resemble the ci/Cro family of repressors found in temperate phages. Both types of repressors prevent excision and replication of the mobile element (PICs and phages, respectively) by binding to specific regions through a N-terminal HTH DNA binding domain^{3,13}. Canonical repressors of the ci/Cro family fuses to this HTH domain a C-terminal domain, of reduced size in several cases, that promotes dimerization¹⁴. Induction of the genes under the repression of these regulators should involve the disruption of the dimeric organization by the interaction with a derepressor protein or by the direct cleavage between the N- and C-terminal domains mediated by proteases in the bacterial host cell (such as RecA*, induced by the SOS response)^{15,16}.”*

Line 174 - For the model – The authors need a methods section explaining how they generated this model. Also energy minimization should be performed on this model before any structural analysis. The authors analysis of this model includes a discussion on specific interactions of Stl with its DNA recognition sequence (Stl box sequence/references not shown, referred to again in line 467), but the authors do not explain and illustrate atom to atom interaction between DNA and protein. If they were to keep this analysis, it would be necessary to model in the exact recognition sequence of Stl in order to identify if the electrostatic interactions in 2B make sense; they should highlight non-specific vs specific interactions in this analysis and a figure highlighting the location of the Stl box recognition sequence in respect to the dimer on their model to illustrate that the dimer “length” matches the length of the recognition sequence. Thus as it stands the conclusion in Line 468 is invalid. The Stl Box sequence and references are needed, and it is not sufficient to just say that the dimer “seems to be compatible with Stl box recognition”.

We thank the reviewer for this comment since the generation of the model was not properly explained in the first version of the manuscript. In the new version we have added a Methods section describing how the model was generated. In this section we explain that the model was subjected to energy minimization. Indeed, two different software were used to ensure that the final model was not biased, obtaining almost identical models. The minimized model included the exact DNA sequence recognized by Stl. DNAProDB software was used for analyze the protein-DNA interactions and a figure highlighting specific and non-specific interactions

has been added (Supplementary Figure 5). As can be confirmed in this new figure, all the residues shown in sticks in Figure 2b interact with the DNA. For clarity we have not highlighted these interactions in Figure 2. Following the reviewer indications, references to the paper describing the DNA recognition sequence as well as the sequence have been included in the manuscript and the position of the recognition sequence in the model has been highlighted in Figure 2a. We have also colored the StI molecule as in Figure 1b to be consistent along the manuscript and following the suggestions of the reviewer in other concerns.

All references to “Indirect” interactions with DNA should be changed to “nonspecific” (those interactions with DNA backbone) for example, Line 189 “Indirect read-out”

“Direct” and “Indirect” interactions have been changed to “specific” and “nonspecific”, respectively.

For Fig 2A - completely remove A, the Authors are not focusing on the dimer and this is confusing given they solved it as a monomer (and the dimerization comes up later). Fig 2B could be in the supplement with Figure S2. This figure allows the authors to say that the helix A3 is close to DNA and could be interacting with the major groove, and to test those residues in panels C and D. However, for these point mutations, the Authors are lacking any negative controls (ie residues that are not predicted to interact with the DNA).

Although Figures 2a and 2b show models, we believe these figures are illustrative and important to understand the experiments represented in Figures 2c and 2bD. Therefore, we have kept them in the manuscript (with the modifications indicated previously).

Negative controls corresponding to several mutations in StI residues not predicted to interact with the DNA are shown in Figures 4a, 6b, 6d and 7b. We have indicated this fact in the manuscript.

Lines 213-216- the Authors state that they confirmed the mutants were dimers in solution and that they had the same denaturalization T_M range as WT protein but they only show 2 mutants, and the mutants do not match WT in SEC. The SEC needs to be shown for every mutant, and since the S44A is of concern - they should only be testing mutants that better recapitulate the WT protein, more like the R51A.

The molecular weight and thermostability of the mutants have been calculated and the data is now reported in Supplementary Table 2. This data confirms that the mutations have no major effect in StI dimerization and stability.

Figure 2C – has extraneous information and missing controls. The beta-gal assays in the presence of Duts are irrelevant, the point is the EMSAs show DNA binding is inhibited, and uninduced beta-gal assays WT vs mutant are consistent with loss of repression. However, to show that these proteins are indeed functional they should be showing interaction with Duts via native page for example. This would confirm that their proteins are not just misfolding and therefore not binding DNA

We agree that the beta-gal assays in the presence of Duts are irrelevant. We have now repeated the experiment using a non-lysogenic strain as a host.

As we indicated in our previous response, all of these mutants are dimers in solution and present similar thermostability profiles to that of the wild type protein (new Supplementary Table 2), supporting that the proteins are well folded. Therefore, respectfully, we believe that to introduce the interactions experiments would not significantly improve the manuscript.

Instead, we have undertaken the reviewer's suggestion from a subsequent comment to simplify the data that we are presenting in order to make it more accessible to readers.

Line 229 - 'complex formation induces reduced changes' not clear what reduced changes are.

The changes induced are explained in the fourth sentence following this one. Changes, which are more important in Stl, correspond to the relative rotation of domains. The complete explanation of the changes induced by the formation of the complex is now described as: *"Comparison of the Bovl-Stl^{N-ter} Dut ϕ 11 complex with the structures of both proteins alone showed that complex formation induces modest changes, mainly in Stl (Supplementary Fig. 6), indicating that their structural conformations in solution are competent for interaction. For Dut ϕ 11, the free (PDB 4GV8)²⁵ and Stl-bound structures are virtually identical (RMSD of 0.16 Å for the superimposition of 161 residues) (Supplementary Fig. 6a). On the other hand, the Bovl-Stl^{N-ter} shows a small rotational movement (around 18 degrees, calculated with DynDom 3D²⁷) between the HTH and middle domains, in which the short α 6 acts as a hinge (Supplementary Fig. 6b), supporting some independence between both domains. Individual comparison of each Bovl-Stl^{N-ter} domain in the free and bound structures showed that the HTH domain (RMSD of 1.19 Å for 55 residues) suffers a higher number of local re-organizations than the middle domain (RMSD of 0.58 Å for 60 residues), which mainly affects the DNA recognition helix α 3 and the following loop connecting α 4 that is partially disordered."*

Mention in line 225 solved by molecular replacement (using what? Include PDB ID) and they didn't mention how the N-terminal and C-terminal structures were solved (in Figure 1).

We used the structure of Dut ϕ 11 alone (PDB 4GV8) as model to solve the structure of the complex. The PDB ID of the model is now indicated in this sentence. In any case, the Methods section includes a detailed description of the protocol followed to solve the structures and the PDB used when molecular replacement was chosen as technical approach.

Figure 3A – The color scheme can be refocused for better clarity- such as monochrome for the Duts, and keep the color scheme of Stl from the previous figure (HTH/Middle etc), there needs to be a focus on how Dut binding leads to inhibition of Stl binding to DNA, they focus more on the function of Dut and how Stl mitigates Dut activity. Hence again why the introduction needs to expand more upon how Dut activity is blocked by Stl. Also for this figure, where is the Dut-Stl interface relevant to the function of Stl? Where is the Stl DNA binding interface in relation to the Stl-Dut interface?

We have completely modified the Figure 3 to keep the color scheme of Stl as in the previous figures. We understand the reviewer's complaint and we thank her/him for this comment since by conserving the scheme color the readers can easily localize the position of the Stl functional domains, particularly the DNA binding domain. Colors for Duts have been changed to different tones of yellow-orange for clarity. In Figure 3a we show the structure of Bovl-Stl^{N-ter}-Dut ϕ 11 complex, as indicated in the title of the figure legend. We do not try to explain how Dut precludes STL binding to DNA. In the Figure we only try to illustrate the Dut-Stl interaction and what structural elements are used for this interaction. The relevance of these interactions in the context of the activity of each protein (enzymatic activity and DNA binding for Dut and Stl, respectively) is discussed in the text of the manuscript.

Figure 3B/C: The motifs need to be labelled more clearly (with different colors) as I can't tell which are which. It would be helpful to have a supplementary figure of the trimeric Dut with the Motifs labeled (Motifs referred to in line 245)

Following the reviewer's comments, the motifs have been now highlighted using different colors. An additional supplementary figure has been added (Supplementary Figure 1) labeling these motifs in the structures and sequences of both dimeric and trimeric Duts.

Figure 3C and line 254 - they don't actually show mimicry, this looks like it is actually just occluding the nucleotide binding site, the tyrosine is not overlaid perfectly so is not mimicry, all wording/claims of mimicry here and similar claims elsewhere should be removed (line 276; in reference to Figure S7 the authors again discuss mimicry, however, the data presented does not prove mimicry (do the residues match, or do the residues actually interact? Would need to see hydrogen bonds or van der waals interactions).

We apologize if our explanation of mimicry was not clear and could lead to confusion. The structures show that almost all the Dut residues involved in interactions with the substrate dUTP in the Dut-dUTP complexes also mediate interactions with StI in the Dut-StI complexes. In this way, StI was "mimicking" the substrate. This observation does not mean that there is a perfect structural mimicry, and that residues of StI occupy spatial positions identical to the functional groups of the dUTP, but that StI mediates interactions that mimic those provided by the substrate. Now, we have clarified this point by indicating in the text that StI mimics the "interactions" mediated by the substrate. We have also included a supplementary table (Supplementary Table 4) listing the residues of the Duts that interact with the substrate and if these same residues interact with StI in Dut-StI structures, supporting this mimicry of interactions.

Figure 4A - labelled Y112113A should be re-labelled StIYYAA. The take home for this should be simplified, the important finding is that the double mutation doesn't respond to trimeric Duts, but doesn't stop the response to Dimeric, don't show the point mutations that didn't work here. In general the authors need to simplify the data that they are showing to make it accessible to readers.

Following the reviewer's comment, we have re-labelled the figure. These mutations were performed because the structural data suggested these residues are important. We think it is important to show the point mutations that do not work to provide a detailed map of the relevant interactions that occurs in the different complexes.

Lines 319-324 Based on the southern blot data shown, it not convincing that there is absolute selectivity of the trimeric vs dimeric, ie the dimeric D1 Dut is not inducing the StIYYAA. In the last western for StIYYAA- the NM1 Dut is running lower than it did in previous panels, these should also be normalized in some way to reflect uneven expression of Duts.

We really appreciate the careful analysis that the reviewer has made related to this figure. In response to the reviewer's comment, we have now prepared a new figure 4B addressing all the points made by the reviewer.

In addition, to complete the *in vitro* analyses we have now analysed the transfer of the different SaPI_{bov1} mutants *in vivo*. As can be seen in the new Supplementary Table 5, the transfer of the island encoding the StI YY-AA mutant was severely reduced after induction of phages $\phi 11$ or 80α (encoding a trimeric Dut), but was completely normal after induction of phage $\phi NM1$ (encoding a trimeric Dut). Also in agreement of the Southern blot analyses, the transfer of the island encoding the StI Y112A mutant was normal for phages $\phi 11$ and $\phi NM1$, however this mutation abolished island transfer by trimeric Dut-encoding phage 80α .

In figure S6, the authors show that the StIY112A is still inhibitory to phi11 with trimeric DUT, even though in Figure 4b, they show that this point mutation results in no SaPI excision/replication with expression of phi11 Dut. By spot assay, the YYAA and Y112A should be identical, and they are not? Further, the spot assay in Figure S6 should be done with D1, since they show that it still interacts by Native gel in 4C, but in 4B southern blot looks like the SaPI is not de-repressed.

All these discrepancies have been now amended in the new versions of the figures. As can be seen now in the Southern blot analyses, in the spot assays, and with the transfer *in vivo* of the islands, induction of the YY-AA mutant is affected for both trimeric Dut-encoding phages, while the loss of induction ability also extends to the Y112A mutation for phage 80 α .

Unfortunately, phages O11 and DI can't be tested in the spot assays because they do not produce clear plaques in the recipient strains. This explanation is now included in the text.

Figure 5 - line 367 - since referencing the surface, it would be helpful to show a surface view of each protein, with the surface highlighted.

We have modified the text to be more accurate with the figure referred.

Lines 361-400 - discussion here is way too long and does not actually reflect what is being shown in the figures. For example, In lines 376-380 - no contacts are actually shown the figure, they are just listing residues that appear in proximity, but there is no indication of what the rubric is for deciding an interaction. It appears as though residues are near each other and the authors conclude mimicry, this is not justified.

The surface of interaction and the protein-protein contacts were analyzed with PISA, CONTACT (from CCP4 suite) and PDBsum software. These programs give a list of the interactions based on distance, geometry (important for hydrogen bonds and salt bridges) and the nature of the atoms that mediate these interactions. The intermolecular StI^{C-ter} - StI^{C-ter} interactions are listing in the supplementary Table 1 and Dut ϕ O11-StI^{C-ter} interactions are listing in Supplementary Table 3. The intermolecular Dut ϕ O11 has been previously reported (Donderis et al. PLoS Pathogens, 2017). Therefore, the interactions are available for the readers, but since we are comparing 3 complexes (2 homocomplexes and 1 heterocomplex) several pages are needed, making the analysis difficult for the reader. Instead, we decided to process the data and show it in a single graphical figure (Supplementary Figure 9) where it is highlighted which residues make contacts. If the reader needs more detailed information about a specific contact can check the supplementary tables 1 and 3 tables or, alternatively, the PDBs that would be available when the manuscript will be published.

A description of how the interactions have been analysed is included in the Methods section of the revised manuscript.

If the latch is mimicking the Dut dimers own latch, is the Dut active as a dimer? How then does the latch differ from the Dut latch (as it must, or Dut wouldn't be active), again suggests not mimicry. In line 384 - they say that it occupies the position of the nucleotide but then say mimicking several of its interactions, but this isn't shown: in S8 indeed H233 doesn't even look like it is in the same position. Rather than mimicking it looks much more like competitive binding. The discussion of the magnesium in S8 in line 393, there isn't a tetrad in the figure so how are we supposed to see mimicry?

Yes, the Dimeric Duts are active as dimers. The Dut latch of one monomer covers the nucleotide-binding site of the neighbor monomer and interacts with the dUTP placed in this active site. In the case of the Stl, the latch is inserted in the Dut active center, mimicking the interaction “of the substrate” with the Dut, but not “mimicking” the interactions of the Dut latch with the substrate or with the neighbor monomer of Dut. We apologize for our error in the writing that could lead to confusion. We have reformulated the sentence so that it is clear that what is mimicked are the interactions that mediate the substrate and not the Dut latch: *“However, this BovI-Stl “latch” is inserted into the active center of Dut ϕ O11 and occupies the position of the nucleotide (Fig. 5c). In this position, the BovI-Stl latch mimics several of the dUTP interactions with the Dut ϕ O11 (Fig. 5c; Supplementary Table 4 and 6).”*

We agree with the reviewer that Figures S8 and 5C do not show correctly what is indicated in the text. We have modified Figure 5c showing the nucleotide to be observed as Stl mimicking the interactions mediated by this substrate. For clarity we have not included lines to show the interactions. These interactions are described in the new Supplementary Table 4. Supplementary figure S8 has been eliminated due to redundancy.

Figure 6A the interaction between trimeric Dut - there is a change in banding pattern to Figure 4C (where 2 bands form indicating the interaction, 6A with WT Stl is not convincing despite the asterisk), is this a reliable technique? How many times were the assays performed? Are replicates going to be shown in the supplement?

Native-PAGE is a very reliable and consistent technique. We have performed these assays several times (for some samples more than 6 times) with identical results. The discrepancies observed are due to the acrylamide percentage used in the gels and/or the running time. We have amended the discrepancies in the figures using gels with similar acrylamide percentage and running these gels similar time, showing the gels now with similar banding in both figures.

Line 413 - why would you expect only a minor effect, shouldn't it have no effect? The gel in Fig 6B is made of 2 different gels that are pieced together, since the control for the gel is on the far left (WT without Dut) this should be repeated. Looks sort of like the C-term mutations make binding to the trimeric Dut tighter, and therefore more free DNA, is this true?

We agree with the reviewer, if trimeric Duts have no interactions with the Stl C-terminal portion, mutations in this part shouldn't have an effect in the trimeric Dut-Stl interaction. We have corrected the sentence. We have also modified Figure 6B showing now two independent gels with the corresponding controls. Differences in free DNA between replicates are minimal, thus we cannot conclude that mutations in C-terminal make binding to trimeric Duts tighter.

Line 421 - referring to aromatic ring, this should be a separate figure.

The stack of Y234 aromatic ring between K238 and M242 can be observed in Figure 5C, thus it does not require a separate figure.

Line 469 - what is an important clash, need to show clash analysis (software available).

The clashes were analyzed with contacts and the results are now indicated in the materials and methods section as follows *“Analysis with Contact³⁶ showed that no clashes between subunits in the dimer occur (shortest distance between side-chains was 2.65 Å).”*

Line 470 - The authors state that dimerization interface is provided by helix 5, however the N

term segment was a monomer as revealed by their SEC and structures, so how can this be concluded?

We apologize for the wrong explanation. We wanted to state that in the model the helix 5 is placed at the dimer interface and should provide interactions in dimerization. As the reviewer comments, these interactions shouldn't be enough to stabilize the dimer, since the N-terminal portion is monomeric in solution. Consequently, we have reformulated the sentences as follow to reflect this fact more accurately: *"The BovI-StI is reported to be a dimer in solution and we have confirmed this by SEC analysis (Supplementary Fig. 4). However, the BovI-StI^{N-ter} was monomeric in solution (Supplementary Fig. 4), which is surprising since several 434 cro repressors and R-M controllers form dimers in solution mainly through the interaction of the HTH and the following helix $\alpha 5^{14}$."* In the section, we concluded that this helix is involved in dimerization from the results of the crosslinking and mutational assays, which were designed from the model.

Cannot do the analysis in Figure 7A without first minimizing the model as mentioned above also show the hydrogen bonds and any salt bridges as dashes on the structure.

As we have indicated in previous answers, the model has been minimized. For clarity, we prefer to describe the interactions in the text and not indicate hydrogen bonds and salt bridges as dashes on the structures.

The band in the EMSA in figure 7B StIH188C without Dut looks off, perhaps that mutant is defective in DNA binding and it shouldn't be included in the analysis.

The mutant conserves the DNA binding capacity. We have included a new gel where the binding is unambiguously observed.

Minor points:

Legend for Fig 4 -B and C are mixed up.

We have corrected the mix up.

Lines 241-244 can be removed, not informative.

We cannot agree with the reviewer. We believe they are informative, since they indicate that complex formation does not stabilize these parts of the proteins. We would prefer to retain these lines.

Line 410 - missing a word between that maintains (should be that StIGGGS maintains the interaction)

Typos:

Line 589 - has be able to

592 - side-directed

Line 529 - replace hypothetical belief with hypothesis

Line 379, R235→R253.

The sentence and the typos have been corrected. We thank the reviewer for her/his careful review of the manuscript.

Reviewer #2:

The manuscript by Marine and colleagues described the means by which phage proteins can directly bind to and de-repress a master regulator in *S. aureus* in order to propagate the content of a pathogenicity island. Prior studies from this group showed that binding of sequence and structure divergent inducer protein proteins to the a master repressor results in island induction. Here, the investigators have used a structure-based approach to analyze the details of how divergent proteins (specifically dUTPases) can be enraged by the same repressor (bov1) to relieve repression. The work is nicely done but the style of presentation is really difficult. Constant use of jargon, and insufficient background details in both the Abstract and Introduction thwart what is an otherwise very elegant study. The authors are advised to consider that the readership of Nature Comm. extends beyond the microbiological community, so a manuscript that is more tailored to a broader scientific audience should be the goal.

Following the reviewers' comments we have substantially modified the introduction section trying to avoid the use of jargon and including the maximum possible background. We hope that now the introduction will be easily understandable and more informative, making it more accessible to Nature Communications readers.

A second major concern is that much of the work is focused on how depression is mediated by disrupting dimer formation. However, no quantitative details for any of the binding proteins is provided. Even a simple analysis using quantitative native gels would provide a much stronger supporter for their hypothesis.

In previous manuscripts, which are referred to herein, we have provided quantitative information about trimeric Duts-Stl interaction and we have quantified the effect in this binding of mutations in Dut catalytic and structural residues (Maiques et al, Nucleic Acids Res 2016; Alite et al, Sci Rep, 2017). Following the reviewer's comment we have now quantified the interaction of dimeric Dut with Stl. In addition, we have also quantified the affinity of Stl mutants with affected dimerization by Biolayer interferometry (trimeric Duts) or MicroScale-Thermophoresis (dimeric Duts, since these Duts were reluctant to be measured by biolayer interferometry, the technique used in our previous studies with trimeric Duts). The results, presented in a new table (Table 2) support the conclusion of the manuscript. We thank the reviewer for her/his comments, which have lead us to carry out these experiments, clearly improving our manuscript.

If the authors are able to address each of these concerns, this manuscript will be sufficiently above the high bar for a journal such as Nature Comm.

Minor points:

1. Figures 3C and %c should show an omit map around the residue that mimic interactions at the dUTP-binding pocket.

Figure 3c and 5c are complex since both show the Dut-Stl interactions. Incorporation of omit maps will difficult the observation of the interacting residues and the position occupied by the nucleotide (represented in sticks). Therefore, for clarity, we prefer to introduce a new supplementary figure (supplementary Figure 13) showing the electron density of the corresponding omit maps around the residues highlighted in Figure 3c and 5c.

2. Table 1 should include the Molprobitly clash score. Also, what is "others" in 3rd line above Ramachadran refer to? There should be some clarification here.

Following the indications of the reviewer the MolProbity Clashscore has been included in Table I. We have also clarified that “others” correspond to ions (sulfate, Mg and Ni) and ethylene glycol molecules present in the crystals.

REVIEWERS' COMMENTS:

Reviewer #1 (Remarks to the Author):

The revised manuscript by Ciges-Tomas is much improved, both in the clarity of the writing and presentation of data. There are several minor suggestions below that I recommend be incorporated prior to publication

- 1) The title does not read well, specifically 'deciphering structurally' and 'evolutionary strategy'. Deciphering structurally sounds grammatically incorrect (though it is correct, it doesn't read well), and as a matter of opinion I don't think the 'evolutionary strategy' (as pertaining to the authors' assertion that 'domain recruiting' is happening) is a strong enough point of this paper to be included in the title. I recommend a shortened title that is focused on the major point of the paper.
- 2) Line 65 – explain what str is
- 3) Line 69 - "Different SaPIs encode different StI repressor proteins, which are highly divergent in sequence, meaning that each SaPI requires a different phage inducer protein for island induction."
 - a. Reword this, as it is not necessarily true and also because in line 78 they prove this exact point. Just because it has a different sequence doesn't mean it has a different binding partner (and this would need a citation anyway to claim this). Maybe say "... which would suggest that each SaPI requires a different phage inducer protein for island induction. However..."
- 4) Line 81 – it is assumed that the recombinases are unrelated structurally, this sentence should be removed as does not add anything concrete.
- 5) Line 109: "...high levels of transferability and indications have been shown that this" rephrase - ...high levels of transferability and this mechanism could help...
- 6) Line 118: "fuses to this" This is too much of an active phrase. This is a better term for describing engineering constructs, but not when nature does it. Maybe "is linked to the C terminal HTH domain..."
- 7) Line 143: "inducers employ different strategies"... too much of an active phrase, as if they want to induce a PICI. Could just take out "or does each of the.....induce the island" part , because that is a given from the first part of the question.
- 8) Line 256: "using as model the structure" change to "using a model of the structure"
- 9) Line 639: "why we did not find" change to "why did we not find"
- 10) I still disagree with the domain recruiting idea put forward by the authors as the wording indicates an active, purposeful process which I think is highly misleading.

Reviewer #2 (Remarks to the Author):

The authors have been able to address each of the points raised by the two reviewers and I believe that this has significantly improved the quality of the manuscript. I am happy to endorse publication in Nature Comms. pending two small minor changes:

1. Please round off unit cell dimensions to the tenth place (i.e. 77.4, 37.3).
2. Screw axes in space group designations should be subscripts or indicated in brackets so as to avoid confusion.

REVIEWERS' COMMENTS:

Reviewer #1 (Remarks to the Author):

The revised manuscript by Ciges-Tomas is much improved, both in the clarity of the writing and presentation of data. There are several minor suggestions below that I recommend be incorporated prior to publication.

1) The title does not read well, specifically 'deciphering structurally' and 'evolutionary strategy'. Deciphering structurally sounds grammatically incorrect (though it is correct, it doesn't read well), and as a matter of opinion I don't think the 'evolutionary strategy' (as pertaining to the authors' assertion that 'domain recruiting' is happening) is a strong enough point of this paper to be included in the title. I recommend a shortened title that is focused on the major point of the paper.

We have changed the title to: The structure of a polygamous repressor reveals how phage-inducible chromosomal islands spread in nature.

2) Line 65 – explain what str is

Str is the SaPI transcription rightward regulator. This information is now included in the text.

3) Line 69 - "Different SaPIs encode different StI repressor proteins, which are highly divergent in sequence, meaning that each SaPI requires a different phage inducer protein for island induction."

a. Reword this, as it is not necessarily true and also because in line 78 they prove this exact point. Just because it has a different sequence doesn't mean it has a different binding partner (and this would need a citation anyway to claim this). Maybe say "... which would suggest that each SaPI requires a different phage inducer protein for island induction. However..."

Following the indications of the reviewers we have modified the sentence. Now read: "Different SaPIs encode different StI repressor proteins, which are highly divergent in sequence, suggesting that each SaPI requires a different phage inducer protein for island induction."

4) Line 81 – it is assumed that the recombinases are unrelated structurally, this sentence should be removed as does not add anything concrete.

The sentence has been removed.

5) Line 109: "...high levels of transferability and indications have been shown that this" rephrase - "...high levels of transferability and this mechanism could help..."

We have removed this part of the introduction since it is redundant a reduction in the text is required.

6) Line 118: "fuses to this" This is too much of an active phrase. This is a better term for describing engineering constructs, but not when nature does it. Maybe "is linked to the C terminal HTH domain..."

We have modified the sentence as follow: "Canonical repressors of the cl/Cro family link to this HTH domain a C-terminal domain, of reduced size in several cases, that promotes dimerization¹²"

7) Line 143: "inducers employ different strategies"... too much of an active phrase, as if

they want to induce a PICI. Could just take out “or does each of the.....induce the island” part, because that is a given from the first part of the question.

We have modified the sentence to avoid the redundancy. The sentence now reads: “for a specific repressor, is this mechanism conserved among the different inducers?”

8) Line 256: “using as model the structure” change to “using a model of the structure”

We have corrected the sentence: “The structure of BovI-Stl^{N-ter} in complex with the trimeric Dut of phage ϕ 11 (Dut ϕ 11) was solved to 2.52 Å resolution by molecular replacement using the structure of Dut ϕ 11 as a model (PDB 4GV8²²; Table I).”

9) Line 639: “why we did not find” change to “why did we not find”

We have deleted this part of the introduction since it is redundant a reduction in the text is required.

10) I still disagree with the domain recruiting idea put forward by the authors as the wording indicates an active, purposeful process which I think is highly misleading.

Given the high number of repressor of cl/Cro family with a single domain, the presence of three domains in Stl seems more feasible with a recruiting process, thus we favor this hypothesis. In any case, we have suggested this as a working hypothesis in the discussion section “Our results call into question the potential for success of such a strategy by supporting the idea that, instead of changing the Stl repressor in response to an anti-repressor substitution by the phage, SaPIs can expand the range of their repressors by recruiting domains able to recognize the new phage-encoded anti-repressors, which incidentally perform the same function for the phage”.

Reviewer #2 (Remarks to the Author):

The authors have been able to address each of the points raised by the two reviewers and I believe that this has significantly improved the quality of the manuscript. I am happy to endorse publication in Nature Comms. pending two small minor changes:

1. Please round off unit cell dimensions to the tenth place (i.e. 77.4, 37.3)

We have modified the table in accordance with the reviewer’s request.

2. Screw axes in space group designations should be subscripts or indicated in brackets so as to avoid confusion.

We thank the reviewer for this suggestion. The space group was incorrect in the table and we have now modified it.